# Functional roles of Aves class-specific *cis*-regulatory elements on macroevolution of bird-specific features

Ryohei Seki[1,2,*], Cai Li[3,4,5,*], Qi Fang[3,4], Shinichi Hayashi[2,6], Shiro Egawa[2], Jiang Hu[3], Luohao Xu[3], Hailin Pan[3,4], Mao Kondo[2], Tomohiko Sato[2], Haruka Matsubara[2], Namiko Kamiyama[2], Keiichi Kitajima[2], Daisuke Saito[2,7], Yang Liu[3], M. Thomas P. Gilbert[5,8], Qi Zhou[9], Xing Xu[10], Toshihiko Shiroishi[1], Naoki Irie[11], Koji Tamura[2] & Guojie Zhang[3,4,12]

Unlike microevolutionary processes, little is known about the genetic basis of macroevolutionary processes. One of these magnificent examples is the transition from non-avian dinosaurs to birds that has created numerous evolutionary innovations such as self-powered flight and its associated wings with flight feathers. By analysing 48 bird genomes, we identified millions of avian-specific highly conserved elements (ASHCEs) that predominantly (>99%) reside in non-coding regions. Many ASHCEs show differential histone modifications that may participate in regulation of limb development. Comparative embryonic gene expression analyses across tetrapod species suggest ASHCE-associated genes have unique roles in developing avian limbs. In particular, we demonstrate how the ASHCE driven avian-specific expression of gene *Sim1* driven by ASHCE may be associated with the evolution and development of flight feathers. Together, these findings demonstrate regulatory roles of ASHCEs in the creation of avian-specific traits, and further highlight the importance of *cis*-regulatory rewiring during macroevolutionary changes.

[1] Mammalian Genetics Laboratory, Genetic Strains Research Center, National Institute of Genetics, 1111 Yata, Mishima, Shizuoka 411-8540, Japan. [2] Department of Developmental Biology and Neurosciences, Graduate School of Life Sciences, Tohoku University, Aobayama 6-3, Aoba-ku, Sendai 980-8578, Japan. [3] State Key Laboratory of Genetic Resources and Evolution, Kunming Institute of Zoology, Chinese Academy of Sciences, Kunming 650223, China. [4] China National GeneBank, BGI-Shenzhen, Shenzhen 518083, China. [5] Centre for GeoGenetics, Natural History Museum of Denmark, University of Copenhagen, Copenhagen 1350, Denmark. [6] Department of Genetics, Cell Biology and Development, University of Minnesota, 321 Church Street SE, Minneapolis, Minnesota 55455, USA. [7] Frontier Research Institute for Interdisciplinary Sciences (FRIS), Tohoku University, Aobayama 6-3, Aoba-ku, Sendai 980-8578, Japan. [8] Norwegian University of Science and Technology, University Museum, N-7491 Trondheim, Norway. [9] Department of Integrative Biology University of California, Berkeley, California 94720, USA. [10] Key Laboratory of Vertebrate Evolution and Human Origins, Institute of Vertebrate Paleontology and Paleoanthropology, Chinese Academy of Sciences, Beijing 100044, China. [11] Department of Biological Sciences, Graduate School of Science, University of Tokyo, 7-3-1 Hongo, Bunkyo-ku, Tokyo 113-0033, Japan. [12] Centre for Social Evolution, Department of Biology, Universitetsparken 15, University of Copenhagen, Copenhagen 2100, Denmark. * These authors contributed equally to this work. Correspondence and requests for materials should be addressed to N.I. (email: irie@bs.s.u-tokyo.ac.jp) or to K.T. (email: tam@m.tohoku.ac.jp) or to G.Z. (email: zhanggj@genomics.cn).

It has been argued for several decades that the phenotypic variations within and between species can be established by modification of *cis*-regulatory elements, which can alter the tempo and mode of gene expression[1]. Nevertheless, we still have little knowledge about the genetic basis of macroevolutionary transitions that produced the phenotypic novelties that led to the great leap of evolution and adaptation to new environment. Although numerous efforts have been made to study the evolutionary roles of newly evolved genes in a limited numbers of model species[2], little is known about how the genetic changes underlying the major transitions occurred in the deep time, and how they were maintained through long-term macroevolution.

Birds represent the most recently evolved class of vertebrates, characterized by many specialized functional, physiological and ecological traits, including self-powered flight, bipedality and endothermy[3,4]. Numerous avian traits and their precursory stages are evident in their theropod dinosaur ancestors and maintained to the bird lineage, as seen in their capacity for flight (such as the large stiff pennaceous flight feathers, air-sacs, three-digit forelimbs that developed into wings, pneumatic bones and others)[4–7]. Despite extensive paleontological and anatomical studies on birds and their near close relatives, the genetic background for these emerging specializations remains unclear. The underlying genes and/or their *cis*-regulatory elements are expected to be maintained by strong selective constraints throughout the avian class, and show distinctive differences from other non-avian species.

One possible expectation from the conserved functions is a high level of sequence conservation over long evolutionary timeframes as reported in other animal groups[8,9]. Comparative genomics provides a powerful tool for identifying these elements[8,9]. For instance, genomic comparison across 29 mammalian species revealed that over 5% of the human genome consists of highly constrained sequences across all vertebrates, including a large number of previously uncharacterized functional elements[9]. As new genomes are released across an increasing diversity of species, the lineage-specific spectrum of highly conserved elements can be ascertained, and should offer new possibilities for understanding the evolutionary and developmental mechanisms for lineage-specific traits under adaptation.

Here we identify avian specific highly conserved elements (ASHCEs) by comparing the genomes of 48 avian species that represent the evolutionary history and diversity of extant birds, against a broad sampling of other vertebrate species. Remarkably, >99% of the ASHCEs are located in non-coding regions, and appear to be enriched with regulatory functions. Through further characterization using functional assessment, expression studies, and large-scale *in situ* comparative embryonic expression analysis for genes with most highly conserved ASHCEs, we identify several candidate genes functionally linked with bird-specific traits appear in the limbs. Furthermore, we provide evidence that supports a role for *Sim1* in the evolution of flight feathers.

## Results

**Rare gene innovation but many gains of regulatory elements.** Although there are more than 10,500 extant species of birds, overall the class exibits smaller and more compact genomes than any other vertebrate class, and birds have furthermore lost thousands of the protein-coding genes in their common ancestors after the split from other reptiles[10,11]. By applying gene family clustering analyses across multiple genomes representing birds, mammals, fishes and other reptiles, we found that bird genomes have on average a relatively lower number of paralogous copies within protein-coding gene families than other vertebrates

(Fig. 1a). This result implies that innovation of protein-coding genes might not play a large role in the processes underlying the transitions from dinosaur to the bird lineage. As an alternate explanation, King and Wilson[12] proposed the regulatory hypothesis, in which gene regulation may play an important role in species evolution. We directly tested this hypothesis at the macroevolution level by examining the genomic regions under strong purifying selection across all birds, and investigated if large part of these conserved sequences have regulatory roles, and have essential role in shaping avian-specific traits, including morphological features.

To identify genomic elements that possess specific functions for birds, we constructed a multiple-way whole-genome alignment for 48 avian and 9 non-avian vertebrate species spanning reptile, mammal, amphibian and fish. The 48 birds represent nearly all extant avian orders, and >100 million years of evolutionary history[11]. Using the phylogenetic hidden Markov model[8], we identify over 1.44 million highly conserved elements (HCEs) among the avian genomes, that are at least 20 bp in length and that are evolving significantly slower than would be expected under neutral evolution. We then determined which of these HCEs had either no orthologues in non-avian outgroups (Type I), or whose orthologues exhibit significantly higher substitution rates among outgroups (Type II; Fig. 1b). The genomic regions displaying such unique conservation pattern among birds are subsequently defined as avian-specific HCEs (ASHCEs). In total, our analysis predicted 265,984 ASHCEs ($\geq 20$ bp), representing nearly 1% of the avian genome (*ca.* 11 Mbp in total, Supplementary Table 3). These ASHCEs had an extremely low substitution rate (about 0.0004 substitutions per site per million years, Supplementary Table 7), ~fivefold lower than the whole-genome average.

The highly conserved nature of ASHCEs suggests that mutations within them may have deleterious consequences. If so, we would also expect a lower polymorphism rate in ASHCE regions. We tested this by assessing the sequence polymorphism pattern within chicken populations[13], and indeed, the frequency of SNPs in ASHCEs (1.27 SNPs per kb) is more than two times lower than the whole-genome average (2.59 SNPs per kb; $\chi^2$-test, $P < 2.2e-16$; Supplementary Table 8), and is comparable to the average level in coding regions (1.29 SNPs per kb). These results therefore suggest ASHCEs have not only been under strong selective constraints during long-term avian evolution, but also at the recent intra-specific level.

**ASHCEs predominate in non-coding regulatory regions.** The preferential targets of strong purifying selection are usually on protein-coding regions[9], for example, 17.55% of HCEs lie within coding regions, some three-fold higher than the percentage of coding regions in whole genome (Fig. 1c). We were therefore surprised to observe, that the proportion of ASHCEs that lie within coding regions was *ca.* 50-fold lower (0.31%, Fig. 1c). The predominance of non-coding sequences within lineage-specific HCEs seems to be a distinguishing feature for the avian lineage, as mammalian-specific HCEs identified using the same method consist of a higher fraction of coding sequences (4.1%; Supplementary Table 5). This result corroborates the above observation that very few lineage-specific genes emerged in the avian genome, suggesting changes in non-coding regulatory sequences might play a more important role in the emergence of avian evolutionary innovations than the acquisition of novel protein-coding genes. It provides strong evidence to support the recent hypothesis that the principal evolutionary changes might be governed by complex non-coding regulatory networks[14].

Consistent with the expectation that ASHCEs might contain regulatory elements, we found that about 58% of the ASHCEs

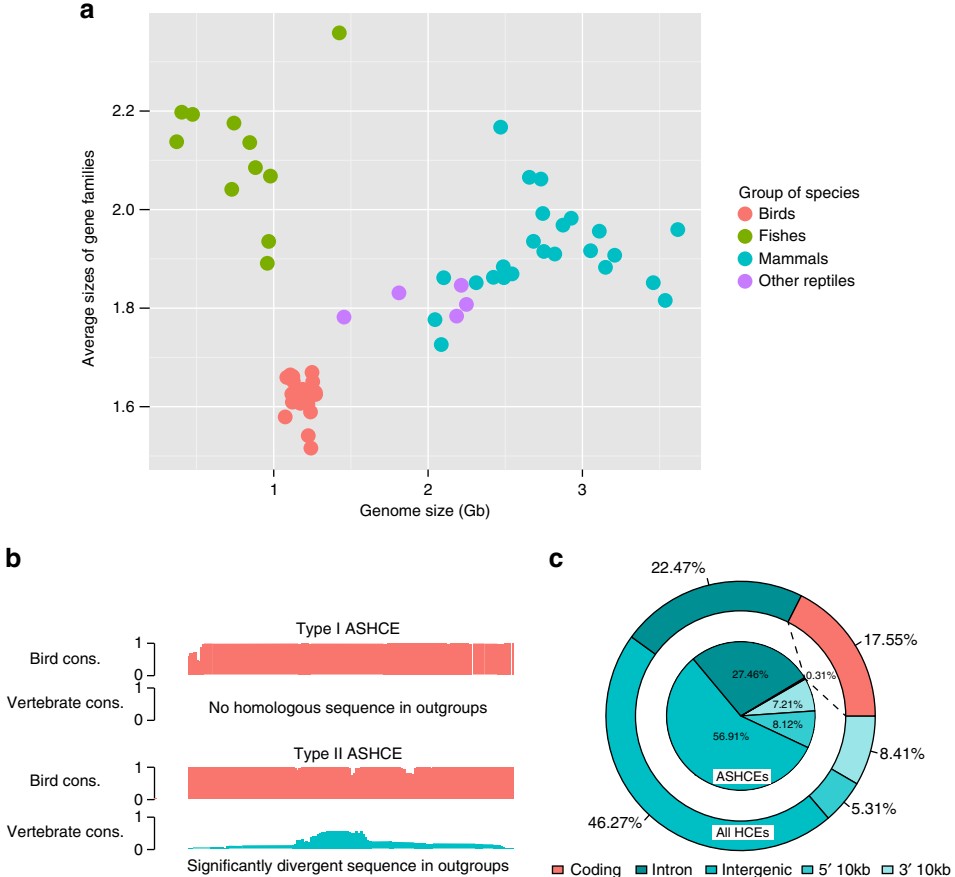

**Figure 1 | Characterization of ASHCEs. (a)** Average sizes of gene families in different vertebrate species. Species in different phylogenetic groups were coloured in different colours. See Supplementary Table 1 for the list of species used in this analysis. (**b**) Illustration of two types of ASHCEs. The type I ASHCEs are conserved in birds but have no homologous sequence in other vertebrate outgroups; the type II ASHCEs are conserved in birds and have rudimentary homologous sequences in outgroup species, but the sequence conservation is significantly low to be detected as homologous between birds and other vertebrates. See Methods section for more details about the identification method of ASHCEs. (**c**) Functional classification of all HCEs and ASHCEs in chicken genome.

contained at least one putative transcription factor binding site (TFBS). What is more, 99 TFBS motifs are statistically over-represented (Supplementary Table 9) in ASHCEs in comparison with their genome-wide background level, and 23 of the corresponding transcription factors were predicted to be involved in regulation of developmental processes (Supplementary Table 9). For example, the transcription factor *Sox2*, which is important for the maintenance of pluripotency in epiblast and embryonic stem cells in mouse[15], has a significantly higher number of binding motifs in ASHCEs than expected ($Q$ value $< 0.05$, calculated by GAT; Supplementary Table 9).

Furthermore, analysis of chicken transcriptome data indicated that 1.62 Mb (14.8%) of ASHCEs were transcribed in at least one tissue, and computational analysis identified 5,511 stable secondary structures among these ASHCEs (Supplementary Table 10). We also hypothesized that some of these ASHCEs might function as non-coding RNAs involved in regulation of their host gene expression. Subsequent investigation revealed that 25 long non-coding RNAs (lncRNAs) overlap with ASHCEs, and two of these lncRNAs are differentially expressed during chicken embryonic development (Supplementary Fig. 1). In addition, we found that almost half of the ASHCEs (43.1% in length) lay inside (within an intron) or adjacent to protein-coding regions (within 10 kb upstream/downstream range of a gene), further implying a role as *cis*-regulatory elements.

**Chromatin-state landscape of ASHCEs**. To assess potential involvement of ASHCEs in gene regulation, we performed genome-wide chromatin immunoprecipitation sequencing (ChIP-seq) for histone modification, since previous studies have shown a strong correlation of chromatin states with functional elements, such as promoters, enhancers, and transcribed regions[16,17]. Because H3K4me1, H3K27ac and H3K27me3 histone markers are often reported to associate with regulatory elements[16], we have investigated these modifications in whole embryos at three key embryonic developmental stages in the chicken, including Hamburger and Hamilton stage[18] 16 (HH16), HH21 and HH32, and limb tissues at HH21 and HH32 (Fig. 2a) with two biological replicates. We further identified peaks of the histone markers in each sample based on the mapping results for ChIP-seq reads. Scanning the chicken genome showed that over 25% of ASHCEs were within peaks of at least one of the histone markers (Fig. 2b). This ratio is significantly higher than the ratio of the whole-genome background, suggesting an over-representation of ASHCEs with a regulatory function (Fig. 2b). Overall, all three histone modifications displayed the expected patterns with respect to the transcription start site (TSS) of whole-genome genes, showing histone modification signal peaking with narrow window around TSS of all genes in whole genome (Fig. 2c). Interestingly, ASHCE-associated genes which were defined by genes with ASHCEs within their genic region or

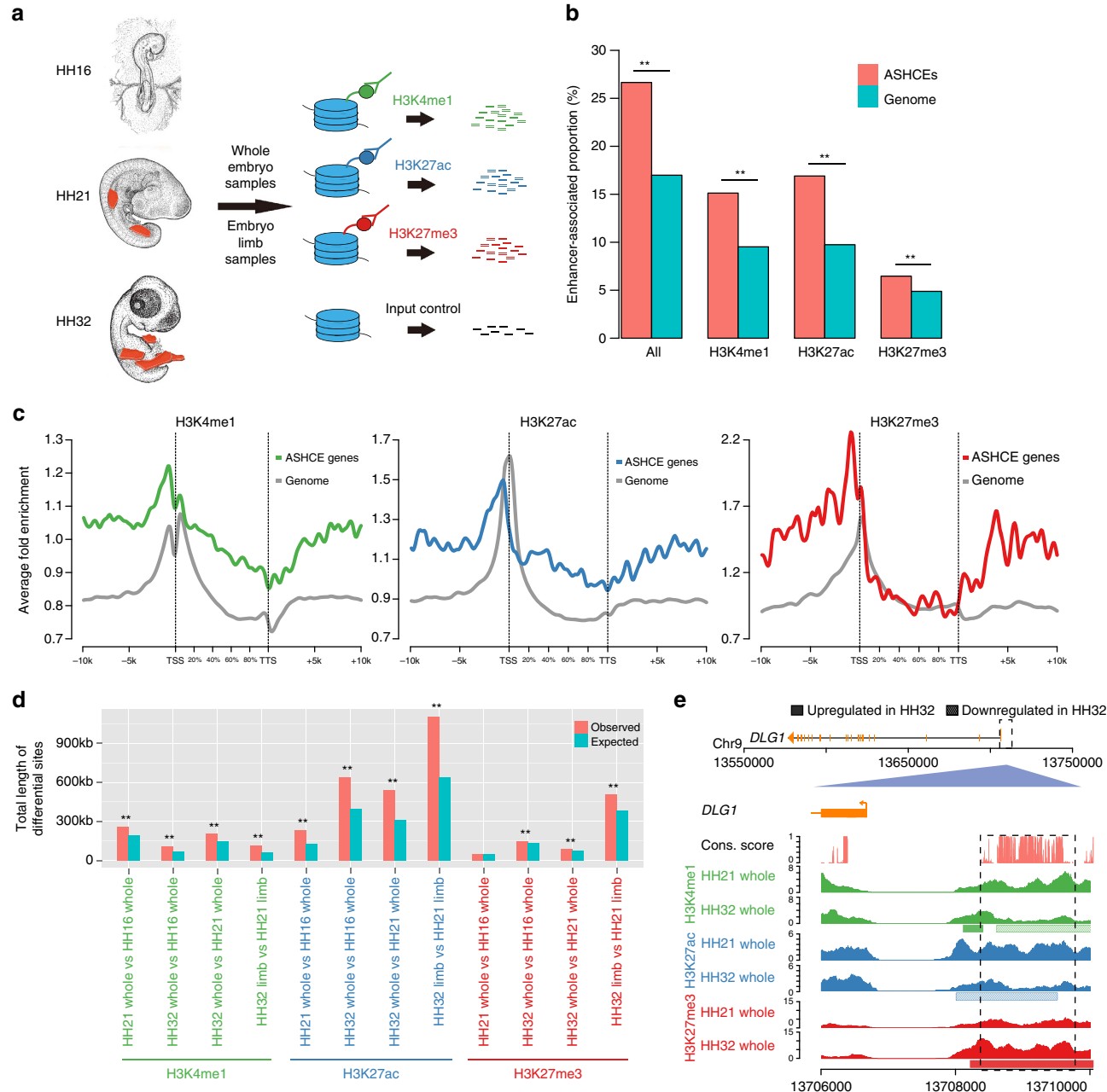

**Figure 2 | Enriched histone marks in ASHCEs.** (**a**) Schematic representation of ChIP-seq experiments. Whole embryo samples from HH16/HH21/HH32 stages and limb samples (red parts on embryos) from HH21/HH32 stages were collected for generating ChIP-seq data of three enhancer-associated histone modification marks (H3K4me1, H3K27ac and H3K27me3). (**b**) Over-representation of enhancer histone marks in ASHCEs. **$P < 0.001$ (calculated by GAT). 'All three marks' means using the non-redundant union set of three types of peaks. (**c**) Average occupancy patterns of histone marks along the genic and 10 kb flanking regions of ASHCE-associated genes and whole-genome genes. The fold enrichment values are normalized density relative to the input samples. The gene body length is aligned by percentage from the transcription start site (TSS) to transcription termination site (TTS). The upstream regions of genes show elevated occupancy relative to gene body and downstream regions. H3K4me1 and H3K27ac of ASHCEs show over-representation relative to genome background in gene body as well as in up- and downstream 10 kb regions, except that the region very near TSS has weak or no over-representation. H3K27me3 also shows over-representation in up- and downstream regions, but not in the gene body. (**d**) Over-representation of differential histone marks in ASHCEs. The differential histone modification sites by comparing two samples at different developmental stages using diffReps. **$P < 0.001$, $0.001 < $*$P < 0.05$ (calculated by GAT). 'Expected' means the expected total length of differential sites if we randomly choose the same amount of loci as ASHCEs from whole genome. (**e**) A case of differential histone modification marks in the upstream of *DLG1*. Normalized fold enrichment signals (normalized with input samples) of three histone marks at HH21 and HH32 stages are shown. The differential regions between HH21 and HH32 predicted by diffReps are also shown (bars under HH32 tracks).

within 10 kb upstream/downstream of the transcription start or termination sites, contain a significantly higher level of overlapping rate with histone modified regions in these embryonic stages compared with the average level of all genes from the whole genome (Fig. 2c, Supplementary Table 17, $P$ value $< 0.05$, calculated by GAT). This suggests regulation of these genes may have become more rigorously controlled and maintained during the development.

To investigate the combinatory patterns of the three types of histone modifications in the chicken genome, we ran chromHMM[19] to generate a four-state chromatin map for each developmental stage by integrating the ChIP-seq profiles of three marks. On the basis of the co-occurance patterns of the four states (Supplementary Fig. 2), we classified them as 'strong enhancer', 'weak enhancer', 'poised enhancer' and 'low signal'[19] (Supplementary Fig. 2, Supplementary Table 18). Of note, the states 'strong enhancer' and 'weak enhancer' are over-represented in ASHCEs in all samples (Supplementary Table 19), further comfirming the regulatory roles of ASHCEs.

Moreover, dynamic changes of histone modification in ASHCEs were observed during chicken development after pharyngular stages (Fig. 2d,e and Supplementary Table 20, $P$ value $<0.05$, calculated by GAT). ASHCE-associated genes are enriched with genomic sites showing different histone modification during development (Fig. 2d), and the same pattern can also be observed when comparing the chromHMM states between different developmental stages (Supplementary Table 21). The change was most dramatic in the H3K27ac marker (Fig. 2d), which is known to be positively correlated with active enhancers[20]. For example, an upstream ASHCE of the gene DLG1, which is found involved in embryo development in mouse[21], exhibits downregulated H3K4me1/H3K27ac and upregulated H3K27me3 at the HH32 stage compared with HH21 (Fig. 2e), suggesting a transition of the underlying regulatory function during development.

We also found some ASHCEs harboured sites that show significantly upregulated histone modification in limb samples than in whole embryo samples (Supplementary Table 22), and many TFBSs were also found over-represented in these regions (Supplementary Table 23). These ASHCEs might be associated with the limb-specific enhancer functions. For instance, of the 16 ASHCE-associated genes, which are assigned with the GO function of 'limb development' (GO:0060173, Supplementary Tables 24,25), we observed ASHCEs harhoring significantly upregulated H3K27ac signal in the limb samples relative to the whole embryo samples in five genes (FMN1, GLI3, LEF1, MEOX2 and PRRX1) (Supplementary Fig. 3). This implies that these ASHCEs may contribute to the regulation of limb development.

**Functional roles of ASHCE-genes in development.** We then investigated the potential function of these ASHCE-associated genes by generating a ranked list of candidate genes based on the conservation level of each gene's associated ASHCEs, and subjected them to statistical Gene Ontology enrichment analysis. Here we found that the top 500 ASHCE-associated genes were enriched in many of the functional categories related to development (Fig. 3a; Supplementary Table 24). This enrichment can also be seen for several categories relating to developmental functions, even when we restricted the analysis to the top 100 genes (Supplementary Table 27). These significant GO terms include embryo development (false discovery rate (FDR)-adjusted $P$ value $= 2.44\mathrm{e}^{-12}$, $\chi^2$-test), head development (FDR-adjusted $P$ value $= 1.29\mathrm{e}^{-05}$, $\chi^2$-test), and limb development (FDR-adjusted $P$ value $= 8.63\mathrm{e}^{-05}$, $\chi^2$-test; Supplementary Table 24). We assessed their conservation level using the non-synonymous/synonymous substitution rate (dN/dS). We found that they were under stronger purifying selection than other genes as they had a significantly lower dN/dS ratio ($P$ value $<0.05$, Wilcoxon rank-sum test, Supplementary Table 28).

We next examined the potential involvement of ASHCE-associated genes in the development of bird-specific features by analysing stage-specific gene expressions at eight early-to-late chicken embryonic developmental stages[22,23]. We found

significantly enriched numbers of ASHCE-associated genes expressed when many avian-specific features become evident, namely, at stages HH28 and HH38 ($P = 0.0005$ for HH28, and $P = 0.015$ for HH38 using Fisher's exact test, Fig. 3b). This is consistent with the prediction deduced from the recently supported hourglass model; genes involved in features related to phylogenetic clades smaller than phylum appear earlier and later than the conserved organogenesis stage, or phylotypic period (stage that is considered to define the basic body plan for each animal phylum)[22,23]. Gata3 and Grin2b were one of these examples, which showed increased expression after the phylotypic period in chicken (see Supplementary Table 34 for more of these genes). We further explored ASHCE-associated genes that were potentially involved in developing avian-specific features. By selecting genes that show more than a fivefold change in expression level after (either at HH28 or HH38) the phylotypic period in chicken (HH16), and further excluding genes that have orthologous counterparts in turtle (Pelodiscus sinensis) and show similar expression change ($>$twofold changes) in turtle embryogenesis, we found 13 ASHCE-associated genes that might be candidates with a specific function in chicken development (Supplementary Table 34). These genes include Gata3, Wnt4 and Grin2b, all of which are reported to be functional components in embryonic development. While these genes show a significantly higher level of sequence conservation between birds and turtle (measured by dN/dS ratio, Wilcoxon rank-sum test, $P = 0.0398$) than other genes, the primary between clade difference of these genes was the presence of ASHCEs in birds that might alter their expression patterns. Together, these results suggest possible involvement of ASHCEs-associated genes in developing avian-specific features. However, as the whole embryonic RNA-seq dataset we used in these analyses lacks anatomical information (for example, where genes are expressed), we thus decided to further investigate the role of ASHCE-associated genes at the tissue levels.

**Comparative expression analysis using in situ hybridization.** To validate how the changes of expression of ASHCE-associated genes have contributed to the development of avian-specific features, we compared cross-species embryonic gene expression using in situ hybridization assays, with a particular focus on developing limbs. In addition to the feathers on the forelimb that are essential to avian flight, bird limbs have other avian-characteristic features that include pneumatized bones, anterior three-digits with short digit 1 in the wing, a wide range of motion in the wrist, a parasagittal gait of the hindlimb, and long metacarpal/metatarsals[24–26]. We thus examined the expression pattern of each of 100 top-ranked ASHCE-associated genes (sorted with the ASHCE phastCons log-odd score) in the developing chicken limb bud at four different stages (from initiation stage of limb development to differentiation stage for limb structures such as the muscle, cartilage and feather; HH20–22, 24–25, 27–29, and 31–33; see Supplementary Fig. 4 for details). Of the 100 genes from ASHCEs-gene list, 30 showed clear and localized expression in the chicken limb bud.

We then carried out a comparative analysis by examining the expression pattern of these 30 genes in the mouse embryo at comparable developmental stages (E10.0, E11.0, E12.5 and E13.5), and found that 10 of these genes showed different expression patterns between the chicken and the mouse (Supplementary Fig. 5). To confirm whether the expression pattern of these 10 genes is unique to birds, we further examined the expression pattern of the above-identified 10 genes in the gecko embryo (Supplementary Fig. 6), and identified four candidate genes (Inadl, Boc, Pax9 and Sim1) that exhibited an avian-specific

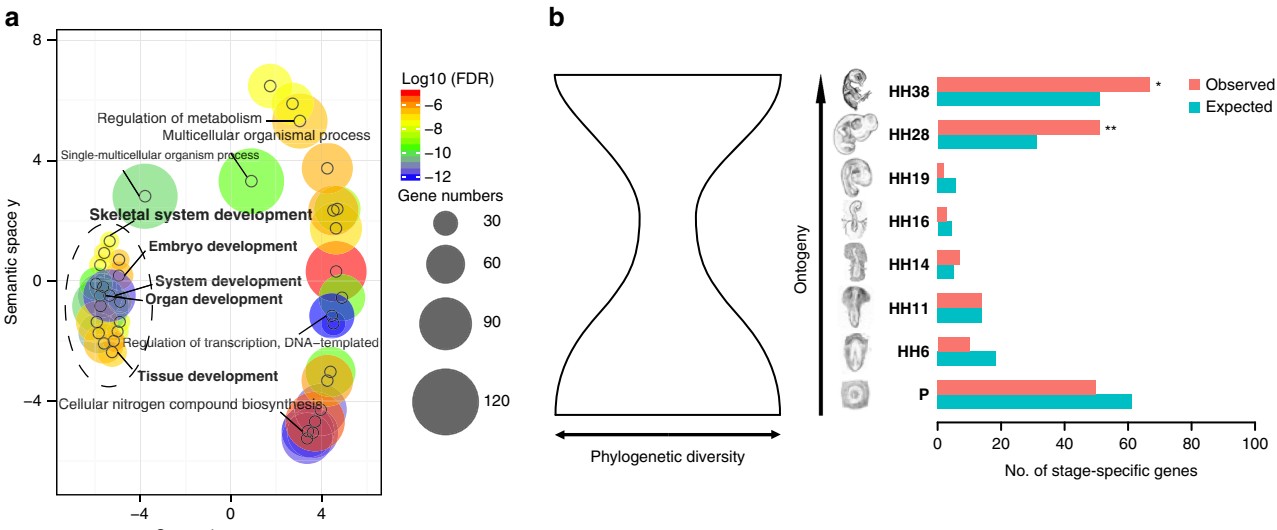

**Figure 3 | Potential role of ASHCE-associated genes in avian development.** (**a**) Enriched GO terms in the top 500 ASHCE-associated genes. The *P*-values of enrichment were calculated using a chi-squared test, and FDRs were computed to adjust for multiple testing. Since the list of enriched GOs was long, the figure was generated by the visualization tool REVIGO which clustered the GOs based on semantic similarity. The development-related GOs are highlighted with bold fonts. (**b**) Stage-specific genes within the top 500 ASHCE-associated genes and those of genomic background (all genes) are shown for each developmental stage. Note that HH28 and HH38 stages show statistically significant over-representation of stage-specific, top 500 ASHCE-associated genes relative to genomic background. 'expected', expected numbers of stage-specific genes among randomly picked-up 500 genes. *$P < 0.05$ using Fisher's exact test; **$P < 0.01$. On the left is the hourglass-like development model.

expression pattern in the developing limb. *Inadl* is expressed in the region around digit cartilage, including phalangeal margin, in both fore- and hindlimbs of the chicken embryo at the differentiation stage of phalanges (HH33), but it is not expressed in the corresponding regions of mouse and gecko embryos (Supplementary Fig. 7a). The expression pattern of *Boc* is similar between three species at the early stage (Supplementary Fig. 6g), but the pattern differentiates at a later stage. Its expression is restricted to the anterior side of the second metacarpal in the chicken embryo at HH32, whereas there is no evident expression in the hindlimb (Supplementary Fig. 7b). In contrast, such restricted expression is neither observed in the mouse nor in the gecko limb (Supplementary Fig. 7b). *Pax9* shows a similar expression pattern in the hindlimbs of chicken, mouse and gecko (Supplementary Fig. 6i), but its expression was barely detectable in the chicken forelimb at late embryonic stages (HH29 and HH32); this contrasts with its high expression in the proximal region of forelimb digit 1 of the mouse and gecko (Supplementary Fig. 7c). Correlating with this is that the forelimb digit/metacarpal 1 in the bird are made up of relatively short bones and special feathers (alula) that are essential for providing the lifting force for flight[27,28]. The inactivation of this gene in mice results in many abnormalities, including phenotypes in the limb such as duplication of digit 1 (refs 29,30). The distinct expression patterns of these genes in chicken when compared with mouse and gecko, implies their unique roles in the bird lineage may be regulated by the existing ASHCEs. What exactly their functions are in avian development will be an interesting target for future experimental investigation.

**Sim1 is associated with flight feather development**. The most interesting gene that we identified as a candidate gene for the development of avian-specific features is the transcription factor *Sim1*, whose expression is specifically restricted to the posterior margin of the developing forelimb at a late stage (HH32) of the chicken limb development, and is neither detected at

corresponding regions in the hindlimb of chicken, nor in homologous regions in both fore- and hindlimbs of mouse or gecko (Fig. 4a), whereas similar expression patterns in all three species could be seen around the basal region of the hindlimb (Fig. 4a) and other areas such as the somite and muscle precursors migrating into the limb bud[31] (among three species, Supplementary Fig. 6e). The positional changes of gene expression in chicken forelimbs (wings) indicate that this gene has obtained a new functional role in avian development.

Closer investigation of *Sim1* expression on sections suggested that it is expressed in the posterior margins of the distal stylopod to the autopod, and the posterior margins of digit 1, especially in limb mesenchyme beneath the epidermis (HH35, Fig. 4b). These *Sim1* expression domains encompass the region where flight feathers (remex-type) develop, indicating a potential relationship between *Sim1* expression and flight feather development. To further explore this hypothesis, we compared the expression pattern of *Sim1* with two general marker genes for feather buds, *Shh* and *Bmp7* (refs 32,33). At HH35, *Shh* and *Bmp7* were expressed along with the *Sim1* expression domain at the posterior margin of the wing (Fig. 4c–e). Moreover, in transverse sections at the zeugopod level in HH36 embryo, *Sim1* is exclusively expressed at the ventral side of the feather bud (Fig. 4f,g) of the remex-type feather (white arrowhead in Fig. 4g–i), while *Shh* (only epidermis) and *Bmp7* (both mesenchyme and epidermis) are expressed at the apexes not only of remex-type feather buds but also other types (Fig. 4h,i). In addition, *Sim1* expression was initiated at a stage between HH30 and HH31, the same time when spot-like expression of *Bmp7* was first observed (Fig. 4j). These results show that *Sim1* expression in the wing corresponds spatially and temporally with remex-type feather formation. To further assess the relationship between *Sim1* expression and flight feather formation, we utilized two chicken breeds, Cochin bantam and Brahmas bantam. These breeds have feet that develop bilateral feathers, including asymmetric flight feather type, in posteriorly-biased mannaer (Fig. 4k and Supplementary Fig. 8a) as seen in feathered-feet phenotypes in other chicken breeds and

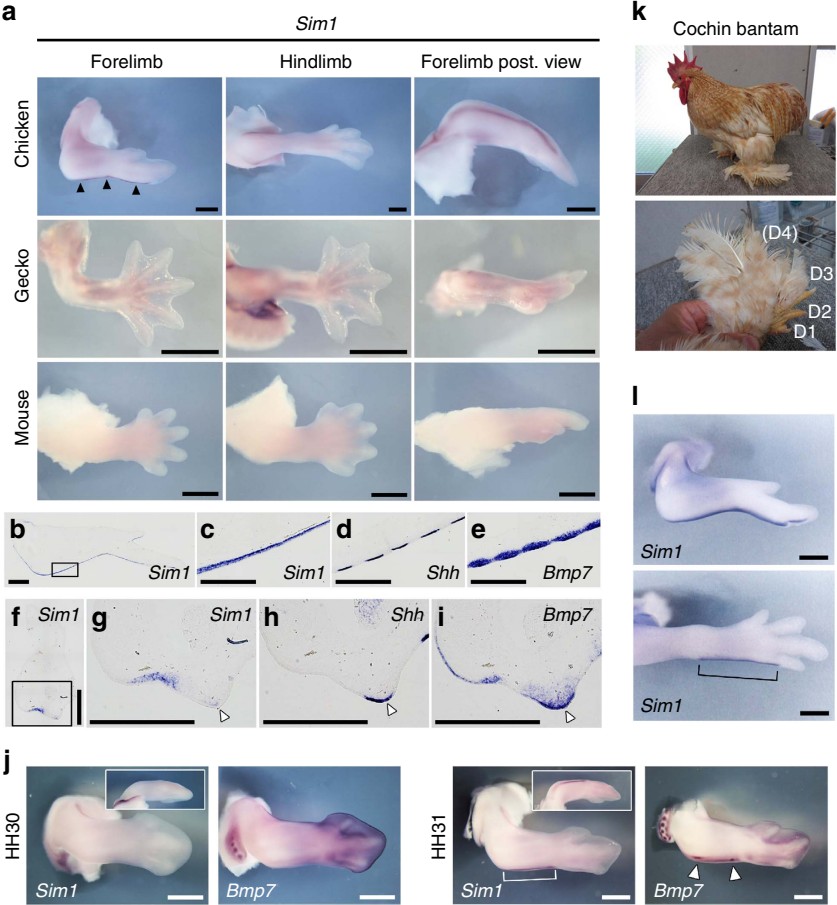

**Figure 4 | *Sim1* is specifically expressed in the avian forelimb and is associated with flight feather development.** (**a**) Expression pattern of *Sim1* in fore- and hindlimbs in chicken (HH32), gecko (23 dpo) and mouse (E13.5) embryos. Black arrowheads indicate specific expression in the posterior margin of the forelimb in chicken. (**b–e**) Expression of *Sim1* (**b,c**), *Shh* (**d**) and *Bmp7* (**e**) on the longitudinal sections of the chicken forelimb at HH35. (**c**) is a higher magnification image of boxed area in **b**. All photographs are oriented with distal to the right and posterior to the bottom. (**f–i**) Expression of *Sim1* (**f,g**), *Shh* (**h**) and *Bmp7* (**i**) on the transverse sections in the zeugopod region of the chicken forelimb at HH36. (**g**) is a higher magnification image of boxed area in **f**. White arrowheads indicate the flight feather buds. All photographs are oriented with dorsal to the right and posterior to the bottom. (**j**) Expression pattern of *Sim1* and *Bmp7* at HH30 and HH31. Neither the *Sim1* expression nor the spot-like expression of *Bmp7* was observed at HH30, whereas both of them were detected at HH31 (white bracket and arrowheads). Insets indicate the *Sim1* expression from the posterior view. (**k**) Photos of the whole body (top) and the feathered foot (bottom) of an adult Cochin bantam. D1-4 incicate digits 1–4, respectively. Digit 4 is not seen from this angle because of heavily covering feathers. (**l**) Expression of *Sim1* in the forelimb (top) and hindlimb (bottom) in the Cochin bantam embryo at HH34. Bracket indicates *Sim1* expression in the hindlimb. Scale bars, 1 mm (**a,b,j,l**); 500 μm (**c–i**).

pigeons[34]. Interestingly, in both breeds we examined, *Sim1* shows clear expression in the posterior margin of the autopod in the hindlimb as seen in the forelimb (Fig. 4l and Supplementary Fig. 8b). We believe the above data strongly suggests that *Sim1* plays a role in flight feather formation.

**An ASHCE enhancer for expressing *Sim1* in the wing.** To elucidate whether the *Sim1*-associated ASHCE serves as a *cis*-regulatory element responsible for avian-specific expression, we conducted reporter assays using two different methods for one of the *Sim1*-associated ASHCEs, that is 284 bp long and is located at the eighth intron of *Sim1* (Fig. 5a). This ASHCE is one of the top 100 significantly conserved ASHCEs. We first used electroporation to test whether this ASHCE could modulate gene expression in the chicken embryo. A 1 kb-fragment sequence containing the ASHCE and flanking regions in both sides was inserted it into a tol2-based reporter vector that contained a thymidine kinase (TK) minimal promoter with an *EGFP* reporter gene (pT2A-TK-*Sim1* ASHCE 1 kb-*EGFP*). The reporter vector was co-transfected into the prospective forelimb field at chicken HH14 embryo by *in ovo* electroporation together with two other vectors: pT2A-CAGGS-*mOrange* for ubiquitous expression as a indicator of the transfection efficiency and pCAGGS-T2TP to express *transposase* for inducing genomic integration (Supplementary Fig. 9a). Six days after electroporation, the reporter EGFP signal was detected in the endogenous expression domain of *Sim1* (Fig. 5b), which was further confirmed in a transverse section (Fig. 5c). A considerable amount of samples exhibited reporter gene expression inside/near the endogenous expression domain of *Sim1* (Supplementary Fig. 10). We also assessed ASHCE activity using a second method, by carrying out RCAN/RCAS retrovirous infection, in which horizontal spread of infection should occur, and thus a broad transfer of vectors would be expected. The reporter vector contains the same 1 kb ASHCE-containing sequence as in the electroporation analysis (Supplementary Fig. 9b). Injection of the retrovirus-infected chicken fibroblasts into the prospective forelimb field at HH10 resulted in expression of the reporter gene in the posterior margin of the wing bud, which was reminiscent of the endogenous

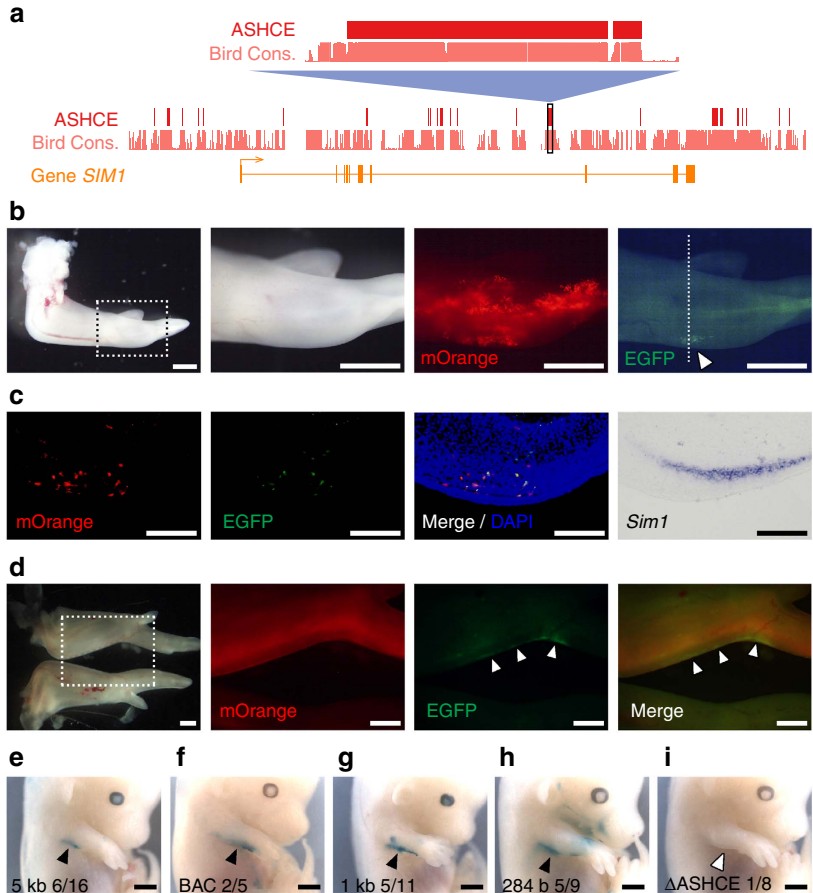

**Figure 5 | A *Sim1*-associated ASHCE represents an enhancer activity.** (**a**) Schematic representation of gene structure of *Sim1*, as well as base-wise conservation scores for ASHCEs. The region harbouring the highest scoring ASHCE associated with *Sim1* is zoomed in. (**b**) Whole-mount images of the forelimb bud six days after electorporation. Ventral views of the right forelimb bud are shown (the original image was flip-flopped horizontally). Dotted box indicates magnified area. The reporter EGFP signals were observed in the posterior edge (white arrowhead) inside broad mOrange signals co-transfected. (**c**) Transverse section on the plane of dotted line in **b**. The signals were detected by immunostaining for EGFP and mOrange proteins. Expression of *Sim1* mRNA on the adjacent section is also shown. Note that the reporter signal was restricted inside/around the endogenous expression domain of *Sim1*. (**d**) Whole-mount images of a specimen 7 days after injection of virus-infected cells. The forelimbs on the top and bottom are the virus-infected and uninfected (contralateral) ones, respectively. Dotted box indicates magnified area, clearly showing that the reporter EGFP signals were detected in the posterior margin of the forelimb bud (white arrowheads). (**e–i**) Reporter expression pattern in transgenic mice. LacZ reporter activities of transgenic embryos (E14.5) of *Sim1* ASHCE 5 kb (**e**), the chicken BAC clone (CH261-127C13) (**f**), *Sim1* ASHCE 1 kb (**g**), *Sim1* ASHCE 284 b (**h**) and *Sim1* ΔASHCE (**i**) were shown. The ratio at the right bottom corner in each (**e–i**) indicates the number of the embryos with the LacZ signal in the posterior margin of the forelimb bud (embryos stained ubiquitously or broadly were excluded), to the number of Tg-positive embryos. Black and white arrowheads indicate obvious and no LacZ signal in the posterior margin of the forelimb, respectively. The reporter vectors used here are shown in Supplementary Fig. 9. Scale bars, 1 mm (**b**,**d–i**); 100 μm (**c**).

expression of *Sim1* (Fig. 5d). These data from two independent analyses strongly suggest that this *Sim1* ASHCE acts as an enhancer in regulation of *Sim1* expression during the development.

To further confirm a regulatory role for the *Sim1*-associated ASHCE, we examined its latent capacity as an enhancer in the mouse embryo by generating transgenic mice. We prepared four types of reporter constructs, all of which contained an hsp68 minimal promoter and the *LacZ* reporter gene, as shown in Supplementary Fig. 9c. The first reporter vector contains chicken *Sim1*-associated ASHCE and its 2.5 kb flanking regions in both sides (*Sim1* ASHCE 5 kb-*LacZ*). The second vector (*Sim1* ASHCE 1 kb-*LacZ*) contains the same sequence used for the reporter assays in the chicken embryo with *Sim1* ASHCE and its shorter flanking sequences. The third vector (*Sim1* ASHCE 284 b-*LacZ*) only contains *Sim1* ASHCE, while the fourth vector (ASHCE negative) lacked the ASHCE completely, but contained its 2.5 kb

flanking regions. When it was introduced, the reporter expression in the first vector could be observed in a posterior-restricted region in mouse forelimb, which partially replicates *Sim1* expression in the chicken wing (Fig. 5e). To exclude the possibility of artificial reporter activity caused by endogenous enhancers around the transgene-inserted site in the mouse genome that can also trigger the expression in forelimb, we modified the chicken BAC clone with *Sim1* by inserting the *LacZ* cassette into the first codon of *Sim1* and generated transgenic mice with this modified BAC clone (Supplementary Fig. 9d). Similar reporter expression was detected in the posterior forelimb of BAC transgenic mice, suggesting this expression pattern should be initiated by the chicken element rather than the position effect (Fig. 5f). The second reporter vector exhibits similar expression to the first reporter (Fig. 5g). The third vector, which only contained the *Sim1* ASHCE, still could activate the reporter gene in the same region (Fig. 5h), indicating that this

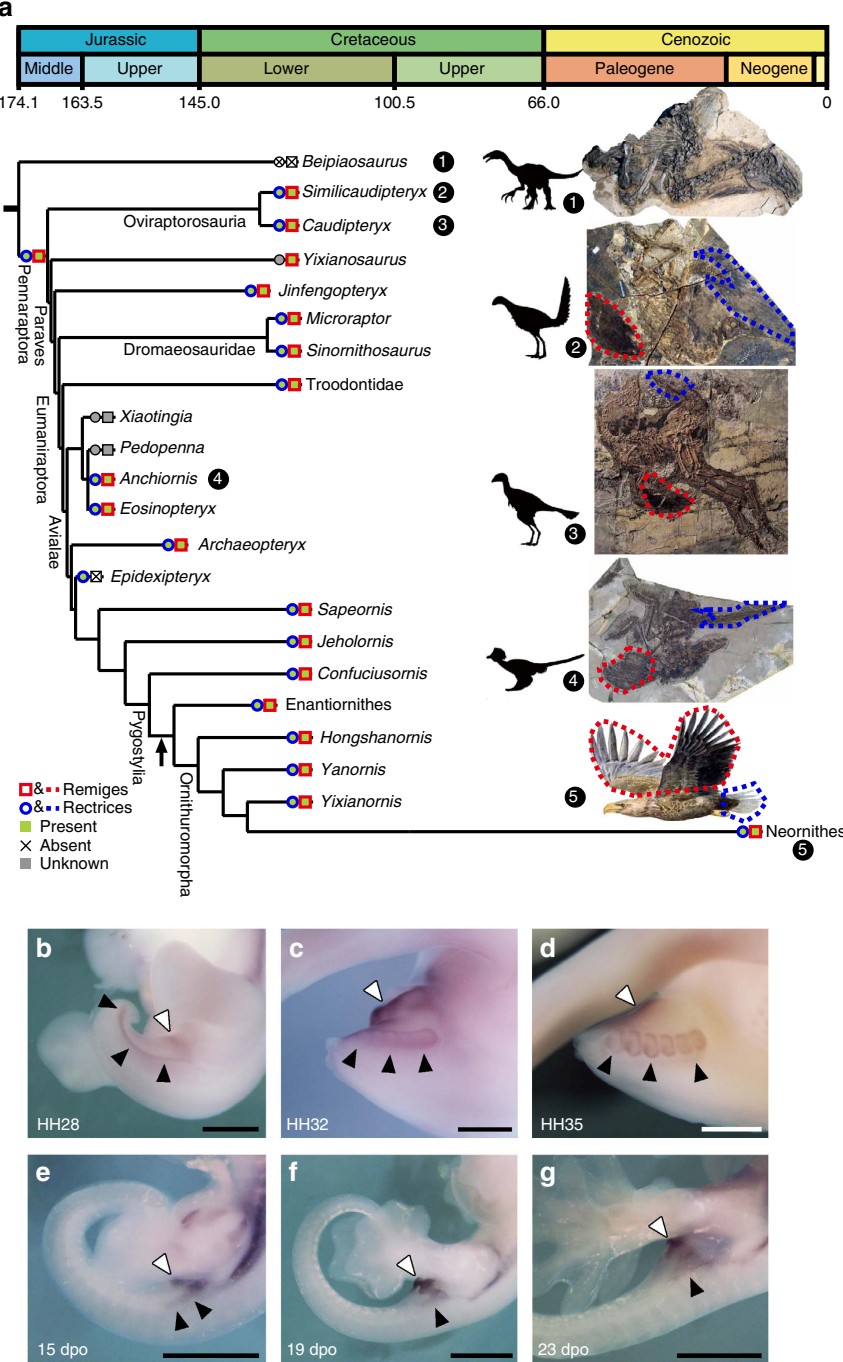

**Figure 6 | *Sim1* and flight feather evolution.** (**a**) Evolution of flight feathers. Character optimization on a calibrated phylogeny indicates that flight feathers (remiges and rectrices) evolved at the base of the Pennaraptora, a clade including oviraptorosaurs, dromaeosaurs, troodontids, and birds, about 170 million years ago. The arrow around the Ornithuromopha branch indicates the estimated time when the *Sim1* ASHCE became conserved (144.0 Myr ago). This tree is modified from the Fig. 3 in Foth *et al*[42]. (**b–d**) The expression of *Sim1* in the tail of chicken embryo at HH28 (**b**), HH32 (**c**) and HH35 (**d**). (**e–g**) The expression of *Sim1* in the tail of gecko embryo at 15 dpo (**e**), 19 dpo (**f**) and 23dpo (**g**). Black and white arrowheads indicate the expression of *Sim1* in the lateral side of the tail and the cloaca, respectively. Scale bars, 1 mm. Drawing of an eagle in **a** is reproduced from ref. 10 with permission. All other drawings are original by the co-authors.

ASHCE contains the full regulatory capacity and is sufficient for induction of gene expression in this region. However, with the ASHCE negative vector, reporter expression was barely detectable (Fig. 5i). These results demonstrate that the 284 bp ASHCE in the *Sim1* locus is sufficient and essential for expressing *Sim1* in the posterior margin of the forelimb. More importantly, our observation that reporter expression is detected in transgenic mice in a manner similar to the endogenous expression of *Sim1* in

chicken wing suggests that the transcriptional machinery essential for activating the *Sim1*-associated ASHCE in the forelimb is conserved among avian and non-avian amniotes, and further implied that rewiring of *Sim1* regulatory network could be achieved by modification on its associated non-coding region. Therefore, it is adequate to postulate that the acquisition of the ASHCE in ancient birds after they diverged from other reptiles should be responsible for the lineage-specific expression pattern

of *Sim1* and the development of corresponding avian-specific feature.

**Sim1 and flight feather evolution.** Flight feathers are one of the most prominent evolutionary innovations in the avian lineage, conferring not only the ability for flight, but also, in some species, important roles in other biological functions, such as territorial displays and courtship ritual[5,35]. Birds have two regions that have flight feathers: along the posterior edges of the wings (remiges) and in the tail (rectrices). Feathered dinosaur fossils[36–39] have provided significant new information on the evolutionary origin of flight feathers. Several analyses of phylogenetic distribution of various feather morphotypes indicate that flight feathers have their origin at the base of the Pennaraptora clade, which includes oviraptorosaurs, dromaeosaurs, troodontids and birds[40–42], and this evolutionary event appears to have occurred about 170 million years ago[7] (Fig. 6a). Given this, we hypothesize that similar genetic mechanisms exist behind the development of flight feathers in wing and tail. If *Sim1* expression is not only involved in modern bird feather development, but also involved in flight feather development in early evolution, we therefore would expect its expression in regions of the tail flight feathers. Consistent with this expectation, we found that *Sim1* was expressed in the both lateral sides of the tail and the region around the cloaca at HH28 chicken embryo (Fig. 6b). At HH32, the buds of flight feathers were observed along the expression of *Sim1* nearby the posterior tip of the tail (Fig. 6c), and its expression was maintained till HH35 (Fig. 6d). In contrast, although *Sim1* expression could also be detected around the cloaca of gecko embryo at 15 dpo, it was only restricted in the region close to the hindlimb, that is, most of the tail region of gecko including its posterior tip did not express *Sim1* (Fig. 6e). Further examination confirmed that the expression in the gecko tail decreased at later stages (Fig. 6f,g). These results suggest that *Sim1* was also expressed in the avian-specific manner at the flight feather-forming region in the tail as well as the wing.

Our molecular dating analysis based on local molecular clocks within the *Sim1* ASHCE alignments indicate that extremely high selection acted on the ASHCE, starting ~$144.0 \pm 26.6$ (s.e.) million years ago (Fig. 6a and Supplementary Fig. 11a–c), a time that is close to the period in which palaeontological data indicate that the first 'modern' flight feathers appeared at the base of the Ornithothoraces, a clade including Enantiornithines and the Neornithes[7,43]. We also highlight that the development of remiges and rectrices has been blocked or inhibited in some modern bird species that lost the ability to fly, such as penguins. Investigation of dN/dS ratio shows there was a much higher *Sim1* gene dN/dS ratio of the penguin branch (dN/dS = 0.2588) than other avian species (average dN/dS = 0.0456, *P* value = 1.77e − 06, Wilcoxon rank-sum test). This may indicate relaxed selection on this gene in penguin lineage, which may be associated with its loss of flight feathers.

## Discussion

It has been a challenge to understand the genetic mechanisms involved in the regulation of the development of lineage-specific traits, particularly for macroevolutionary processes that created a whole class level of new animal group. Previous studies have emphasized duplication and modification of protein-coding gene as a major source of evolutionary novelty and their contribution to the lineage-specific phenotypic evolution[44,45]. In addition, non-coding sequences, including non-coding RNA and *cis*-regulatory sequences, are also suggested to have a great contribution in evolutional change and innovation of traits, but there is little evidence of *cis*-regulatory elements involved in

creation of the class level animal traits. Our study indicates that there are the class-specific regulatory elements that are highly conserved in avian genomes, and further suggests that the majority of genomic elements that are under strong lineage-specific selective constraints in birds consist of non-coding sequences. In comparison with mammals[46], few novel coding-region–based functional elements appear to have arisen in the avian ancestor, suggesting that the development of many avian-specific traits may be through transitions of gene expression profiles by adapting new regulatory networks. Our large-scale functional genomic experiment confirmed that genes associated with ASCHEs are significantly enriched in various development processes, and the ASHCEs may have contributed regulatory functions that could rewire and co-opt the expression patterns of their associated genes. Our study thus provides a dataset of functional candidate regions that would be worth further examination.

We took the identified ASHCE-associated genes and carried out *in situ* analysis of gene expression across species, and provided a potential foundation for inferring the functional regulation of key genes during the development of vertebrate anatomical structures. In particular, our experiments offer evidence for a specific role of *Sim1* in flight feather development and the essential switch on and off regulatory role of its associated ASHCE on its avian-unique expression pattern. The association of *Sim1* expression with remiges and rectrices was accordant with predictions from paleontology data, that these two types of flight feathers evolved simultaneously. This suggests that they might be mediated by a single genetic cascade. The strong sequence conservation of this ASHCE across all bird lineages suggests an adaptive episode in the common ancestor of birds by the formation of this element. We suggest that this evolutionary conservation pattern started with the appearance of the 'modern' flight feathers at Ornithothoraces in the earliest Cretaceous. 'Modern' flight feathers differ from early flight feathers at several salient features, and have been suggested to enhance the flight capability of modern birds[43,47]. The high-selection constraint on ASHCEs might be the consequence of the adaptive transition on flight feather evolution.

More broadly, our study demonstrates a framework for future study to identify genomic modifications responsible for various levels of lineage-specific phenome innovations by integrating analytical tools from genomics, developmental biology, and paleontology. Given our experiments here have only focused on development of a particular organ, the limb, we postulate that there are more functionally relevant ASHCEs that coordinate the regulation of additional avian class-specific features, some of which might have been shared with their ancestors among theropod dinosaurs and served as genomic mechanisms for macroevolution.

## Methods

**Genomic data.** To identify the ASHCEs, we used the 48 avian genome dataset in the avian phylogenomic project[10] and 9 outgroups: three non-avian reptile genomes –*Alligator mississippi*, *Chelonia mydas* and *Anolis carolinensis*; and 6 other representative vertebrates (according to the 7-way alignment in UCSC)—*Homo sapiens*, *Mus musculus*, *Rattus norvegicus*, *Monodelphis domestica*, *Xenopus tropicalis* and *Danio rerio*. For gene family analysis, we chose 22 birds with the highest quality assemblies in the avian phylogenomic project[10], 5 non-avian reptiles, 24 mammals and 11 fishes from UCSC[48], Ensembl and other sources (Supplementary Table 1). To identify the mammal-specific highly conserved elements, we chose 20 mammalian genomes and 8 other vertebrate genomes as outgroups (Supplementary Table 1).

**Gene family sizes and lineage-specific genes.** The protein sequences of four groups (birds, non-avian reptiles, mammals and fishes) were used to run all versus all BLAST alignment and build gene families with the tool *hcluster_sg* in

TreeFam[49]. To compare the gene family sizes across groups, we calculated the average family sizes for the gene families present in at least two groups (Fig. 1a).

**Identification of HCEs.** To identify the conserved elements, we initially generated the pairwise sequence alignments across all avian genomes by LASTZ[50] and chainNet[51] using chicken genome as the reference. We then used MULTIZ[52] to combine the pairwise alignments into multiple sequence alignments. The final alignment contains approximately 400 Mb of each avian genome (Supplementary Table 2). In addition, we also generated the chicken + three-reptiles four-way alignments for three non-avian reptile species against chicken, and downloaded the seven-way whole-genome alignment (chicken as the reference) from the UCSC FTP database[48] for six other vertebrate species. All these alignments were also combined to form a 57-way alignment, which was used for phyloP analysis later.

Firstly, we ran phyloFit in the PHAST package[53] with the topology from avian phylogenomic project[11], to estimate the neutral ('nonconserved') model based on fourfold degenerate sites in the 48 birds alignments. With the nonconserved model as the input, we ran phastCons[8] to estimate conserved models with its intrinsic function, and predicted the HCEs ('--most-conserved' option) in birds and generated base-wise conservation scores ('--score' option). Next we identified two types of avian-specific HCEs (ASHCEs): (1) the HCEs that have no outgroup sequence aligned (Type I ASHCEs); (2) HCEs that have orthologous sequences in one or more outgroups are only conserved in birds according to the phyloP tests in PHAST package (Type II ASHCEs)[53] (Supplementary Table 3). To filter out the non-avian-specific HCEs in Type II ASHCEs, we kept the candidates of Type II ASHCEs with phyloP $P$ value < 0.01 (FDR corrected) in all three separate sets of phyloP tests using three different sets of outgroup: (1) alligator only; (2) three non-avian reptiles; (3) three non-avian reptiles and six other vertebrate species. Because a considerable number of these elements were short, to have a higher quality set of ASHCEs for downstream analyses, we only used the ASHCEs of ≥ 20 bp in subsequent analyses. To do the comparision, we also used a similar method to identify the mammal-specific highly conserved elements (Supplementary Table 4).

To estimate the substitution rates of different branches in the ASHCE loci, we ran phyloFit[53] on alignments of the ASHCE loci with at least one outgroup with a fixed topology (the published TENT tree[11]) (Supplementary Table 7). The divergence times were obtained from previous studies[11,22].

To investigate the SNP density in ASHCEs, all HCEs, coding region and whole genome, we took advantage of the published chicken SNP dataset[13] (Supplementary Table 8).

**Functional analyses of conserved elements.** We used the chicken genome protein-coding gene annotation information in Ensembl to classify the HCEs into five non-overlapping groups: coding, 5′ 10 kb region (relative to the translation start site), 3′ 10 kb region (relative to the translation stop site), intronic, and intergenic. We set a priority order for the five annotation groups as follows: coding > 5′/3′ 10 kb > intronic > intergenic (Supplementary Table 5).

To annotate transcription factor binding sites (TFBSs), we used the JASPAR CORE vertebrates matrices[54] and ran TESS[55] ('-mlo 10 -mxd 5') to predict the putative TFBSs on both strands of chicken genome, and identified the TFBSs located in ASHCEs (Supplementary Table 9). TFBSs predicted by TESS were merged with the motifs predicted with ChIP-seq data (see the ChIP-seq section below) for over-representation analysis.

To investigate expression of ASHCEs, we compared the positions of ASHCEs and assembled transcripts from the chicken RNA-seq data sets[22,56]. To investigate potentially functional RNA structures of ASHCEs, we extracted the multiple alignments of each HCE to run Evofold v1.0 (ref. 57) to identify functional RNA structures (Supplementary Table 10). We compared these RNA structures to the chicken miRNAs annotated in Ensembl (Supplementary Table 11).

To investigate whether the ASHCEs harbour some lncRNAs, we made use of the 6452 annotated lncRNAs[10]. We also performed lncRNA annotation with other RNA-seq data sets[22] using the annotation pipeline described in ref. 10. In total, we obtained the lncRNA collection of 26,749 lncRNA genes. We identified 25 avian-specific lncRNAs overlapping with the ASHCEs with a coverage ratio of > 0.5. Based on avian-specific lncRNAs, we compared the expression levels of chicken phylotypic period (HH16 (ref. 18)) and other developmental stages (Supplementary Fig. 1).

**ChIP-seq analyses of ASHCEs.** We performed ChIP-seq for three known enhancer-associated histone modifications: H3K4me1, H3K27ac and H3K27me3. We collected samples from developing chicken embryos at stages HH16, HH21 and HH32 respectively, and samples from developing limbs at stages HH21 and HH32 respectively. ChIP experiments were performed basically according to the protocol recommended by Cosmo Bio with a slight modification. Modified points are as follows. Embryonic samples were dissected and dispersed by 0.05% trypsin (Gibco, 25300054) for 5 min at 37 °C. After adding an equal amount of FBS to stop the enzyme reaction, the cells were filtered using a cell strainer (BD Falcon, 352360). At least $5 \times 10^6$ cells were used for the subsequent process. The cell lysates were sonicated 20 times (pulsed for 30 s with 30 s interval) using a Bioruptor (Cosmo Bio, UCD-300) at high power setting. Alternatively, we sonicated the

lysates 17 times (pulsed for 15 s with 60 s interval) used a Vibra-Cell (Sonics and Materials, VCX130PB) at 20% amplification setting. The samples were divided into four aliquot and added 10 μg of antibodies (Normal rabbit IgG, Santa Cruz, sc-2027; Anti-Histone H3 mono-methyl lysine 4, abcam, ab8895; Anti-Histone H3 acetylated lysine 27, Active Motif, 39113; anti-Histone H3 tri-methyl lysine 27, Millipore, 07-449). Salmon sperm DNA, which is generally used for blocking, was not applied for all steps.

ChIP DNA samples (two biological replicates for each condition) and input samples were sequenced by Illumina HiSeq2000. The sequencing reads of each sample were aligned to the chicken genome by BWA[58]. After removing PCR duplicates by samtools[59] and removing singletons (defined as reads that did not have any other reads mapped within 100 bases of either side), we chose uniquely mapped reads to assess the signal-to-noise ratios for each sample using the *SPP* package (*R* statistical software package)[60] (Supplementary Table 12).

The input samples of the same condition were merged, and ChIP-seq peaks for each sample were identified by the MACS2 with broad peaks mode ($P$ value < 0.05)[61] (Supplementary Table 13). Furthermore, we calculated the Pearson correlation coefficients between biological replicates (Supplementary Table 14). To obtain the final peak set for each condition, we counted the aligned reads in the peak regions from each of the two replicate samples, and applied quantile normalization on the aligned reads to comparing these values across samples[62]. We only kept reproducible intersected peaks that had an average coverage of ≥ 1 in both replicates for each condition (Supplementary Table 15).

To investigate enrichment of enhancer-related histone marks in ASHCEs, we merged the peaks of a same histone mark from different conditions into a non-redundant union set, and compared the genomic regions of histone marks and ASHCEs using GAT[63] (number of simulations is 100,000, the same below; Supplementary Table 16). We also compared the genomic regions of histone marks and ASHCEs by GAT[63] to see if there is any significant over-representation pattern in the five annotation groups (exonic, 5′ 10 kb region (relative to the transcription start site), 3′ 10 kb region (relative to the transcription stop site), intronic and intergenic, Supplementary Table 17).

To discover motifs (putative TFBSs) with ChIP-seq data, we pooled all ChIP-seq peaks from different samples together to identify significantly enriched motifs using findMotifsGenome.pl in HOMER[64]. We obtained enriched known motifs with a $q$ value smaller than 0.05 and identified *de novo* motifs by default parameters. Using these predicted motifs from the ChIP-seq data as input, we further identified the motif sites in the whole genome with scanMotifGenomeWide.pl in HOMER[64]. Finally, we merged the TFBSs predicted by *TESS* and those by HOMER into a union set, and identified the TFBSs that are over-represented in ASHCEs relative to whole-genome background by GAT (Supplementary Table 9).

We generated a chromatin-state map for each developmental stage by integrating three types of histone makrs using ChromHMM[19]. To determine the suitable number of states in our data, we tested the number of states from 3 to 10 and found that the 4-states results were better than any other when considering the state transition parameters (Supplementary Fig. 2). Based on the co-occurance patterns (Supplementary Fig. 2) and previous literature[19], we defined the four states as 'strong enhancer', 'weak enhancer', 'low signal', and 'poised enhancer'. The statistics of chromatin-state maps are provided in Supplementary Table 18. Over-representation tests for each chromatin state in ASHCEs using GAT revealed that 'weak' and 'strong' enhancers were over-represented in all samples (Supplementary Table 19). We identified the regions with differential chromatin states between two different samples, and found that the differential regions were over-represented in ASHCEs for all comparisons (Supplementary Table 21).

Because of the relatively poor accuracy of the chromHMM state maps with only three marks, we also used diffReps[65] to identify the differential histone modification sites between different conditions for each mark. We tested whether differential histone modification sites were over-represented in the ASHCEs using GAT[63] (Supplementary Table 20). Furthermore, we identified the sites with significantly upregulated histone modification in limb samples relative to whole embryo samples (defined as limb-specific differential sites), and then tested whether these sites were over-represented in the ASHCEs by GAT[63] (Supplementary Table 22). We also indentified the over-represented TFBSs in limb-specific differential sites overlapping ASHCEs (Supplementary Table 23).

**ASHCE-associated genes.** For each HCE, we considered the nearest protein-coding gene as its associated gene, and only considered the genes within 5′/3′ 10 kb range of the HCEs.

To investigate the potential functions of the highest scoring genes related to ASHCE, we first sorted the ASHCE-associated genes by the highest ASHCE phastCons log-odd score in each gene. Subsequently we compiled three kinds of top gene lists for further analyses: top 500, top 200 and top 100. Based on the positions of HCEs relative to the genes, we defined 4 groups further: 1) 'within 10 kb' (including intron/exon/5′ 10 kb/3′ 10 kb); 2) '5′ 10 kb' (located in 5′ 10 kb upstream); 3) '3′ 10 kb' (located in 3′ 10 kb downstream); 4) 'intron'.

We performed GO enrichment analysis ($\chi^2$-tests) for the three top gene lists respectively, using the Ensembl chicken GO annotation[66] (Supplementary Tables 24–27).

We also made use of one-ratio dN/dS estimates of the genes from the avian comparative genomics paper[10]. We compared the one-ratio dN/dS values of the top ASHCE-associated genes with that of other genes using Wilcoxon rank-sum test (Supplementary Table 28).

**Differentially expressed genes (DEGs).** We used published chicken RNA-seq data[22] to identify the differentially expressed genes using DEseq v1.14.0 (ref. 67), edgeR v2.4.6 (ref. 68) and baySeq v1.8.3 (ref. 69). The cutoff P value of 0.01 (FDR adjusted) was used for each method to obtain the initial list of differentially expressed genes (DEGs). The intersection set of the results of the three methods were used for further analyses. For chicken, we compared the expression levels of chicken phylotypic period (HH16) and 7 other embryonic developmental stages (P: Primitive Streak embryos, HH6, HH11, HH14, HH19, HH28 and HH38) (Supplementary Table 29). We applied the same pipeline to identify the differentially expressed lncRNAs during chicken developmental stages (Supplementary Table 30). To identify the stage-specific genes expressed in a specific chicken embryonic stage, we used a measure *tau*[70] (Supplementary Table 31).

To compare the expression levels of chicken and other non-avian outgroups, we performed similar analyses to identify the differentially expressed genes using the RNA-seq data of the soft-shell turtle[22]. For turtle, we compared the expression levels of turtle phylotypic period (Tokita-Kuratani stage 11, TK11 for short) and two late developmental stages (TK15 and TK23, corresponding to HH28 and HH38 in chicken)[22] (Supplementary Table 32). We were interested in the genes differentially expressed in chicken late stages (HH28 and HH38) relative to the phylotypic period, but not differentially expressed in corresponding turtle stages (TK15 and TK23). To reduce false positives, we required > 5-fold changes for chicken DEGs and < twofold changes in turtle non-DEGs (Supplementary Tables 33 and 34). To investigate selective constraint on DEGs, we ran KaKs_calculator[71] on the chicken-turtle 1:1 orthologs from Ensembl[66] to estimate the dN/dS ratios.

**Animals.** Fertilized chicken eggs (*Gallus gallus*, White Leghorn) were purchased from local suppliers and incubated at 38 °C. Embryos were staged according to Hamburger and Hamilton (1951)[18]. Fertilized eggs of two feathered-feet chicken strains (Cochin bantam and Brahmas bantam) were provided by the National BioResource Project (NBRP) Chicken/Quail of the MEXT, Japan. Adult individuals of a male Cochin bantam and a female Brahmas bantam were used for photographs. Foot feathers were obtained from adult male individuals of both strains. Mouse (*Mus musculus*) embryos were collected from pregnant ddY mice that were obtained from a local supplier. Mouse transgenic assays including pronuclear microinjection and recipient mouse husbandry were conducted in National Institute of Genetics, Japan. Gecko (*Paroedura pictus*) embryos were obtained from adult individuals that were kept in KT's laboratory and were staged according to Noro *et al.*[72] (2009). All animal experiments were properly conducted in accordance with the guidelines approved by Tohoku University (2014LsLMO-018, 2016LsLMO-010) and National Institute of Genetics (28-7), Japan.

***In situ* hybridization assays for ASHCEs-associated genes.** We selected 100 genes for *in situ* hybridization from the top 500 'within10kb' genes. As some genes could be poorly annotated, we did not consider the genes with incomplete gene models (no start/stop codons) or containing frameshifts. With regards to all genes that we used for *in situ* comparative embryonic expression screening, partial clones as riboprobes for *in situ* hybridization (except for chicken *Shh* and *Bmp7*, which were kindly gifts from Dr. C Tabin from Harvard Medical School, and Dr. T Nohno from Kawasaki Medical School, respectively) were newly obtained by reverse transcription–PCR and sequenced. Primers used for cloning are listed in Supplementary Table 35. cDNA pools for reverse transcription–PCR were derived from embryos at HH20 or HH27 for chickens, at E10.0 for mice and at 10 dpo for geckos. Templates with Sp6 and T7 promoters for riboprobe synthesis were generated by PCR. All templates were transcribed with an appropriate RNA polymerase. For whole-mount *in situ* hybridization, embryos were fixed in 4% paraformaldehyde at 4 °C overnight, washed with PBT (0.1% tween20 in PBS), dehydrated gradually with a methanol/PBT series, incubated in 5% $H_2O_2$ for 1 h, then stored in methanol at − 20 °C. After gradual rehydration, embryos were digested with proteinase K (Invitrogen, 10 μg ml$^{-1}$ for 10 min), washed in PBT, fixed in 4% paraformaldehyde containing 0.2% glutaraldehyde, then washed in PBT. Embryos were prehybridized at 70 °C for at least 1 h in hybridization buffer (50% formamide, 5 × SSC (pH 5.0), 50 μg ml$^{-1}$ *E. coli* tRNA, 50 μg ml$^{-1}$ heparin, 1% SDS) before hybridization with a DIG-labelled riboprobe (1 μg ml$^{-1}$ in hybridization buffer) at 70 °C overnight. Embryos were washed in solution 1 (50% formamide, 5 × SSC (pH 5.0), 1% SDD) at 70 °C and in solution 3 [50% formamide, 2 × SSC (pH 5.0)] at 65 °C. After washes in TBST (100 mM Tris-HCl (pH 7.5), 150 mM NaCl, 0.1% Tween 20), embryos were incubated in blocking buffer (1.5% blocking reagent, Roche), then incubated in blocking buffer containing anti-DIG antibody (1:2,500, Roche) at 4 °C overnight. Embryos were washed in TBST, followed by incubation in NTMT buffer (100 mM NaCl, 100 mM Tris-HCl (pH 9.5), 50 mM MgCl$_2$, 0.1% Tween 20) before staining reaction with NBT/BCIP. In expression screenings of chicken and mouse embryos

(Supplementary Figs. 4,5), we used around three (at least two) individuals for each gene at each developmental stage. Then, we confirmed that the candidate genes showed almost consistent expression pattern in all specimens we analysed. Regarding the expression analysis in gecko (Supplementary Fig. 6), we used one embryo for each gene and each developmental stage because of a limited number of the fertilized eggs available for this animal. For section *in situ* hybridization, embryos fixed in 4% paraformaldehyde at 4 °C overnight were embedded in OCT compound (Sakura Finetek), then sectioned with cryostat at 10-μm thick on Platinum coated slide grasses (Matsunami). Sections were washed in PBT, incubated in 1 μg ml$^{-1}$ proteinase K at 37 °C for 7 min, then washed in PBT before fixation with 4% paraformaldehyde for 20 min. After washes in PBT, hybridization was carried out in hybridization buffer containing a DIG-labelled riboprobe (1 μg ml$^{-1}$) at 70 °C overnight. Sections were washed in solutions 1 and 3 at 65 °C and then in TBST. After incubation in blocking buffer (0.5% blocking reageant), sections were incubated in blocking buffer containing anti-DIG antibody (1:2,500) at 4 °C overnight. After washes in TBST, sections were incubated in NTMT buffer containing 2 mM levamisole, and finally colour reaction was carried out in NTMT buffer containing NBT/BCIP and 2 mM levamisole.

**Cloning of *Sim1* ASHCE and vector construction.** Approximately 5 kb DNA fragment, including the *Sim1* ASHCE 284 b, was retrieved from the chicken BAC clone (CH261-127C13), which contains the *Sim1* locus (see below for the detailed retrieving method). The other short fragments were cloned by PCR in which the retrieved DNA fragment or male chicken genome (Zyagen, GC-120M) was used as a PCR template. For reporter assay, each DNA fragment was inserted into pT2A-TK-*EGFP*, RCANBP(A) or hsp68-*LacZ* reporter vector (HSF51). Modification of the BAC DNA to insert a *LacZ* reporter cassette into the first ATG of the *Sim1* CDS was performed as follows[73]. The pKD46 plasmid was electroporated into DH10B host *E. coli* carrying the BAC DNA to produce the arabinose-inducible λ red recombinase for homologous recombination. After selection of the cells possessing the pKD46, ∼500 ng of a DNA fragment of the *LacZ* and *Kanamycin* resistance cassette flanked with homology arms (synthesized by PCR) was electroporated into the host *E. coli*. Colony PCR was performed to select a specific recombinant of the BAC DNA. Then, the pCP20 plasmid carrying thermally inducible *flp* gene was electroporated into the host *E. coli* to remove FRT-flanked *kanamycin* gene. Removal of *Kanamycin* was confirmed by colony PCR. To retrieve the approximately 5 kb DNA fragment (*Sim1* ASHCE 5 kb) from the BAC DNA, we partially applied this BAC modification method. Briefly, PCR was carried out to generate ∼300-nucleotide terminal homology arms for the 5 kb DNA fragment. After inserting these homology arms into a retrieving plasmid vector carrying a selection marker gene, we linearized this plasmid by digestion with an appropriate restriction enzyme at the site between the two homology arms and electroporated ∼500 ng of this product into the DH10B host *E. coli* possessing both the BAC DNA and pKD46 plasmid. After overnight incubation, the positive clone was selected by colony PCR. Primers used in these steps are listed in Supplementary Table 36.

***In ovo* electroporation.** Vector plasmids were purified using Qiagen maxiprep kits, and the DNA pellets were resuspended with EB buffer (Qiagen). Each plasmid cocktail was prepared at concentrations of 5–14 μg μl$^{-1}$ in EB buffer, and then coloured with fast green solution for easy visualization. The plasmid cocktail was injected into the coelom in the presumptive right forelimb field at HH 13–14 by using finely pulled grass capillary. Hockey stick-shaped platinum anode and tungsten cathode were put lateral and medial to the right forelimb field, respectively. Electroporation was conducted using CUY21 Vitro-EX or CUY21 EDIT (BEX) under the following conditions: one driving pulse of 25 V, 0.03–0.05 ms pulse length and 1 ms interval length, followed by three poration pulses of 8–12 V, 10–25 ms pulse length and 200–475 ms interval length. Six days after electroporation, the reporter signal was detected by fluorescence microscope. For enhancement of the reporter signal, some samples were fixed and cryosectioned to conduct immunostaining. The antibodies used are as follows: anti-GFP monoclonal antibody (Nacalai tesque, 04404-84), 1:500 dilution; anti-DsRed polyclonal antibody (Clontech, 632496), 1:500 dilution.

**Retrovirus infection.** Two retrovirus vector constructs (RCANBP(A)-*Sim1* ASHCE 284 b-TK-*EGFP* and RCASBP(B)-*H2bmCherry*) were simultaneously transfected to virus-free DF1 cells. After several times of passage, the infected cells were collected and condensed by centrifuge. Subsequently, the condensed cells were injected into the lateral plate mesoderm in the presumptive right forelimb field at HH 10 using finely pulled grass capillary. Embryos were incubated for 7 days, followed by the detection of fluorescent signals.

**Transgenic mouse generation LacZ staining.** Transgenic mice were generated by pronuclear microinjection of a linearized reporter vector. For preparation of reporter vectors, plasmids were digested with appropriate restriction enzymes. Target fragments were cut out from a low melting agarose gel and digested with GELase (Epicentre Technologies) at 43 °C overnight. DNA was purified with phenol and chloroform extraction followed by precipitation with ethanol. Purified DNA was dissolved in injection buffer (5 mM Tris-HCl (pH 7.5), 0.1 mM EDTA

(pH 8.0)). After filtration through 0.22 μm filter unit (Merck Millipore), DNA was diluted in a concentration of 4–5 ng μl$^{-1}$ and used in microinjection. For BAC transgenesis, extracted BAC DNA was digested with PI-SceI (NEB), and the digested fragments were cut out from a 0.8% agarose gel following electrophoresis. After dialysis with a cellulose membrane, the purified BAC DNA (5–6 ng μl$^{-1}$) was microinjected. For LacZ staining, embryos were fixed at E14.5 in 2% paraformaldehyde containing 0.2% glutaraldehyde and 0.2% Nonidet P-40 at 4 °C for 1–2 h. After washes with PBS several times, staining reaction was carried out in PBS containing 0.5 mg ml$^{-1}$ X-gal, 5 mM potassium ferricyanide, 5 mM potassium ferrocyanide, 2 mM MgCl$_2$ and 0.2% Nonidet P-40 at 37 °C overnight[74]. In order to identify transgenic animals, genome DNAs were extracted from the amnions to carry out genotyping PCR using a set of primer pairs for *LacZ*. In the case of BAC transgenesis, we used primer pairs for T7 and SP6 sites in the vector backbone in addition to those for *LacZ*. Primers used for genotyping are listed in Supplementary Table 36.

**Dating analysis of the *Sim1* ASHCE.** Based on multiple sequence alignments of the 284 bp ASHCE in *Sim1* (including sequences from 48 birds, alligator and turtle), we estimated the time point that this element became conserved in birds (more details are provided in the legend of Supplementary Fig. 11), assuming before that time point the element evolved at a neutral rate in birds. The divergence times used for calibration were obtained from previous studies[11,22].

**dN/dS analysis of *Sim1*.** To estimate the evolutionary rate of the gene *Sim1*, we extracted the gene sequences of the 22 birds with the highest quality assemblies. We used PAML[75] to estimate the dN/dS ratio of each branch. To support the penguin ancestral branch has significantly higher dN/dS relative to other birds, we made use of the LRT test based on the two-ratio branch model (one dN/dS for penguin ancestral branch, another dN/dS for other birds) and one-ratio model (one dN/dS estimate for all birds).

**Data availability.** ChIP-seq data that support the findings of this study have been deposited in the GEO of NCBI with the accession code GSE75480. More processed data are provided in the Supplementary Data 1 in the Supplementary Information.

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

## Acknowledgements

We thank Professor Jon Fjeldså of the University of Copenhagen for valuable comments. This project was supported by Strategic Priority Research Program of the Chinese Academy of Sciences (XDB13000000) and Lundbeckfonden grant R190-2014-2827. K.T. was supported by JSPS KAKENHI Grant (JP15H04374), grant from The Naito Foundation, and Next Generation World-Leading Researchers from the Cabinet Office, Government of Japan (LS007). R.S., S.E. and H.M. are JSPS Research Fellows (JSPS KAKENHI Grant Numbers JP14J07050 (R.S.), JP15J06859 (S.E.), JP15J06385 (H.M.)). C.L. was partially supported by Lundbeckfonden grant R52-5062 to M.T.P.G.). N.I. was partially supported by Platform Project for Supporting in Drug Discovery and Life Science Research Platform for Dynamic Approaches to Living System from the Ministry of Education, Culture, Sports, Science and Technology (MEXT) and Japan Agency for Medical Research and Development (AMED). Photographs of the adult Cochin bantam and Brahmas bantam and their fertilized eggs were provided by the National BioResource Project (NBRP) Chicken/Quail of the MEXT, Japan.

## Author contributions

G.Z., K.T. and N.I. designed the study. C.L., Q.F., J.H., L.X., H.P. and Y.L. conducted the computational analyses. R.S., S.H., S.E., M.K., T.S., H.M., N.K., K.K. and D.S. conducted the wet-lab experiments and analysed the data. G.Z., K.T., N.I., X.X., R.S. and C.L. wrote the manuscript. M.T.P.G., Q.Z. and T.S provided critical comments for improving the manuscript.

## Additional information

**Competing financial interests:** The authors declare no competing financial interests.

