## [Peer Review File · Nature Communications]

Reviewers' Comments:

Reviewer #1 (Remarks to the Author)

A Summary of the key results

- Seki et al. identify avian-specific highly conserved elements (ASHCEs) and ask if the genes they are associated with show evidence of being involved in specifying avian-specific morphological features.
- They first identify ASHCEs using a comparative genomic approach and show that they are depleted in coding regions compared to HCEs
- They use chromatin immunoprecipitation experiments, publically available whole embryo chick RNA-seq time-course data, and molecular evolutionary analyses, to guide a large scale comparative in situ hybridization strategy that seeks to discover genes whose expression has been rewired during avian evolution. Their study focused on limb-related phenotypes.
- Results from in situ hybridization analyses highlight several genes that appear to have a distinct expression in chick compared to mouse and gecko.
- The authors focus on one transcription factor Sim1 and show that its associated ASHCE can drive a unique gene expression pattern in both chick and mouse embryos.

B Originality and interest: if not novel, please give references

- This is an original and interesting study. Cotney et al. Cell 2013 (PMID: 23827682) have previously used comparative epigenetic analyses of H3K27ac to discover lineage specific limb enhancers in primates. Infante et al. Dev Cell 2015 (PMID: 26439399) have recently looked for limb-genital enhancer usage in snakes, again relying on H3K27ac to identify such enhancers. However this study appears to be the first genome-scale effort to explore this important question in birds.

C Data & methodology: validity of approach, quality of data, quality of presentation

- Although they gave a link to their data, the authors did not provide a password for the reviewer to access their data. Given the detailed QC and informative summary given by the authors I presumed the data was indeed submitted to GEO. I recommend that the authors consider providing the many processed data files (e.g. entire list of ASHCEs, genome browser tracks and differential enrichment analyses of ChIP-seq data, list of top500 genes), to increase the impact and value of their study to the research community.
- Figure 1 has some minor grammatical issues in figure and legend.
- I find that "47% of ASHCEs contained at least one transcription factor binding site (TFBS)." is not really that compelling. I presume almost any sequence would have at least one predicted TFBS. It may be leveraging the epigenetic datasets collected in this study prior to doing motif enrichments (i.e. do limb enriched H3K27ac regions give any specific motif enrichments?).
- On page 8 the identity of the lincRNAs is not given. Only reference I could find appear to be analysis-specific annotations in the supplemental file (e.g. "CUFF.3981"). I recommend the authors provide a list of the differentially expressed genes used in their analyses.
- On line 169 authors mention they created 'a chromatin state map'. No chromatin state map is mentioned in the methods nor is one shown in Figure 2b. To me chromatin state maps are used to integrate different histone modifications (i.e. PMID: 23221638). Such an analysis of the whole embryo and limb ChIP-seq data could be useful to compare to ASHCEs.
- Summary Lines 173-176 - the results reported for the ChIP-seq experiments appears to be more of a QC analysis. Could the authors classify all their ASHCEs based on some epigenetic criteria (e.g. high H3K27ac, low H3K4me1 = putative promoter; high H3K27ac, low H3K27me3 active enhancer; high H3K27me3, high H3K4me1 latent/poised enhancer)? For example there is no mention of which ASHCEs show evidence of H3K27ac enrichment (in whole embryo or limb). Also the biology of H3K27me3 is not brought into the study. Do any ASHCEs show evidence being

regulated by polycomb complex? Were any H3K4me1+H3K27me3 high and H3K27ac low regions found amongst the ASHCEs? Are there any ASHCEs that are also H3K27ac enriched in limb? If so I would recommend including/discussing these regions in the later in situ experiments. I would argue that an epigenetically active ASHCE in developing chick limb is more biologically relevant to limb development than an ASHCE ranked by the DNA constraint of its closest gene. The H3K27me3 data generated in this study could result in unique discoveries, however it seems to me this data was not effectively used to explore the biology of ASHCEs.

- Figure 3A is cluttered and how the annotations are placed on the figure is not clear. I am not sure this is a helpful addition to the main text of this manuscript.

- On line 187 the authors report: "We also found that some ASHCEs harbored with sites showing significantly up-regulated histone modification in limb samples compared to whole embryo samples. These ASHCEs might be associated with the limb-specific enhancer functions (Supplementary Table 19)." The main text and Table 19 legend seem to imply that there are 564,596 limb-specific enhancers in ASHCEs. However there are only 34,000 limb enhancers in total reported for the HH32 limb ChIP-seq data. This needs to be clarified. It may be the authors are referring to the number of base pairs of ASHCEs that are overlapping H3K27ac enriched peaks. However the text and supplemental figure legend seem to refer to ASHCEs themselves, not individual base pair coverage of ASHCEs. Without access to the processed data I am not able to look further into this. This is crucial as it is essentially the only analysis that the authors perform where individual ASHCEs are compared to the epigenetic data. The previous limb enhancer evolution studies mentioned earlier, underscore the rationale for leveraging epigenetic data in discovering limb enhancers. More effort to integrate/compare the epigenetic results to the DNA constraint-based analyses would likely provide important biological insights.

- The explanation for the 'differential histone modification' in Figure 2e is incomplete. For example which of the 3 histones are differentially enriched (and at what cutoff). It appears there may be a decrease in H3K27ac, H3Kme1 at HH32 and an increase in H3K27me3 relative to HH21. The fact that H3K27me3 is a repressive histone modification is not mentioned in the manuscript. A more thorough analysis of the histone modification data and a clear linking of the results to the existing gene expression data and novel in situ data would be helpful. For example the only example given was at DLG1, however I did not see any discussion of this gene later in the manuscript. I was wondering why it was chosen as opposed to the ASHCEs explored later in the paper? In principle a detailed analysis of the epigenetic data will provide classes of ASHCEs and the authors could provide examples of representative ones. For example what is the epigenetic status of Sim1 in limb (the epigenetic data would fit well as part of Fig 5a for example)? It is of course possible that the avian specific innovations occur in a small number of cells and that bulk ChIP-seq or RNA-seq will not have the power to detect them. If nothing was learned from the epigenetic status of ASHCEs in the developing limb then this needs to be made clear as it would contradict results from previous studies. Such ideas/caveats should be explored in the paper.

- Line 217: clarify which prediction is being referred to.

- It is not clear how the top100/500 list used for the comparative gene expression in situ analyses (line 235) connects to the ChIP-seq experiments. This should be made clear. Based on the previous limb enhancer literature I would be particularly interested in the function of ASHCEs near known limb/genital enhancers. However the sense I get from this paper is that DNA sequence constraint was the primary means of prioritizing limb enhancer innovations.

- Line 255 uses Indl abbreviation and rest of manuscript uses Inadl.

- Although the QC provided by the authors is well described, it is not clear how robust the ChIP-seq experiments are. Is the y axis in Figure 2E some sort of normalized value or does it represent read number. It would be helpful to show input on the same scale. More examples of genome browser tracks for the most important, and representative ASHCEs would be valuable.

- Figures 5B, C and D are somewhat difficult to interpret as the results presented here do not seem to show consistent EGFP expression within the Sim1 endogenous expression domain in the chick embryo. In contrast the mouse reporter data looks more convincing. It seems more work is needed to justify the author's conclusions in the chick system. For example, a previous study that identified Sim1 as a chick and mouse limb enhancer (PMID: 19123137; this paper was not cited in

the current study) mention that they had to resort to doing in situ work sections.

- Supplemental Figure 2 seems to be blurry compared to supplemental Figure 3.

D Appropriate use of statistics and treatment of uncertainties

- No caveats are given regarding the limitations of using of whole embryo RNA-seq data to identify ASHCE-genes in development (line 207).
- How variability in the in situ work was treated was not clear. How many embryos were analysed for each in situ and what criteria was used to select a representative pattern could be made more clear.

E Conclusions: robustness, validity, reliability

- The identification of the ASHCEs appears to be robust.

F Suggested improvements: experiments, data for possible revision

- Gene expression data (qPCR or RNA-seq) that matches the limb development ChIP-seq data would be ideal and could allow authors to better identify genes which are regulated during limb development.
- In their last paragraph of the section "Functional roles of ASHCE-genes in development", the authors identified "13 ASHCE-associated genes" that "might be the candidates with specific function in chicken" from comparison of chicken and turtle. I am curious to know what the expression patterns of these 13 genes in their in-situ analysis. Do they show unique expression pattern in chicken? Are they among the top 100 ASHCE-associated genes?
- (line 205) The most conserved genes (top 500) nearby an ASHCE's were said to be enriched for "limb development". However the genes that gave rise to this enrichment were not mentioned (Table 21 mentions there are 16 of them. Note Table 23 was referenced in place of Table 21 where the enrichment was given). It is unclear if any of the known limb genes that had a nearby ASHCE were explored or highlighted in the ChIP-seq, RNA-seq, or in situ experiments. Is Sim1 one of these known genes? For these known limb genes it would be justified to identify all nearby ASHCEs and not be restricted to only 10kb upstream and down stream. Epigenetic activation in the limb could also be used to prioritize the functional study of these limb related ASHCEs. Given the tremendous effort already put into in situ hybridization it would be worth exploring this angle and perhaps even profiling a few of these if they haven't already been done.
- It is understandable why the authors focused on in situ patterns that were different in chick and mouse. However the recent literature (e.g. Cotney et al. and Infante et al.) seem to suggest to rewiring of existing limb genes has lead to lineage-specific morphological innovations. If it is difficult to assess the quantitative differences of limb gene expression via in situ hybridizations between species, would it not be reasonable to prioritize all limb related genes with a nearby ASHCE rather than using DNA constraint scores for prioritization? For example a previous study that showed Sim1 was a chick limb enhancer concluded the expression patterns were similar between chick and mouse (PMID: 19123137). To me the expression patterns driven by the enhancer associated with TFAP2C seems to be another excellent candidate. The authors have done a huge amount of in situ validations however more discussion of the current literature and the biology behind alternative prioritization schemes (e.g. known limb genes, epigenetic activity as opposed to measures of purifying selection may have been a more appropriate way to identify limb related enhancers.)

G References: appropriate credit to previous work?

- The authors have not acknowledged or incorporated ideas from papers directly related to their work. For example:

- 1) Limb enhancer study of Cotney et al. How does this work relate to the current study?
- 2) Infante et al. Is there evidence of genital expression patterns changing along with limb expression patterns during avian evolution?

3) The association with Sim1 to limb development in chick and mouse seems to have already been published in 2009 (PMID: 19123137). The early work claimed that Sim1 had a similar expression in chick and mouse which is different than the conclusions of this study. A comparison of this previous work to the current work is warranted.

H Clarity and context: lucidity of abstract/summary, appropriateness of abstract, introduction and conclusion

- The context and rationale are compelling and clear. The relationship between different experiments/analyses in the paper could be strengthened as mentioned above.

Reviewer #2 (Remarks to the Author)

Authors ask a great question on the macro-evolutionary process from reptiles to the birds. In an attempt to identify genes responsible for avian-specific features, Seki et al performed a comparison of the 48 avian genomes with 9 non-avian species looking for avian specific highly conserved elements (ASHCEs). They found that the vast majority of ASHCEs were located in cis regulatory non-coding regions rather than producing novel coding genes. Using an extensive in situ hybridization screen covering different developmental stages to identify expression patterns, they identified several genes that correlated with bird-specific limb development, including bone and feather formation. Four of these genes (Inadl, Boc, Pax9 and Sim1) were not present in mouse or gecko limbs and were further characterized. The expression of Sim1 in the chicken limb suggests it may be involved in the acquisition of flight feathers. This result was tested using electroporation of a reporter construct, as well as transduction with a retroviral construct and the use of transgenic mice. Sim1 ASHCE directed reporter expression in the location similar to the endogenous gene. While they have done a body of works that identify many cis-regulatory elements, and increase our knowledge toward the wonderful big question asked, it is not convinced that they have found genes that define "bird specific trait" or define the avian class.

1. Coumailleau and Duprez, *Int. J. Dev. Biol.* 53:149-157 (2009) using section in situ hybridization reports that Sim1 shows similar expression patterns in both mice and chickens at similar stages as described here. They demonstrate that it is involved in early forelimb muscle formation. This paper was not mentioned in the current manuscript. How do the authors reconcile this paper with their own findings?

2. Although they show ASHCE found in Sim1 can direct expression to the specific region of the limb adjacent to where flight feathers form, they don't clearly show that Sim1 is involved in limb or feather development. Functional experiment is essential to establish the causal relationship. If Sim1 is a real flight feather determination gene, overexpression of Sim1 ASHCE or Sim1 gene body in the specific region in hind limb should induce the flight feather formation in the leg. The authors should do this experiment.

3. They should also evaluate the expression of Sim 1 in the hind limbs of birds with feathers on their feet. Shapiro claims that these are flight feathers and authors might expect to see Sim1 expressed with a similar pattern as that seen in forelimbs in these birds as opposed to birds with scales on their feet.

4. Sim1 expressed in both wing feather and tail feather. The expression of Sim1 in these feathers could be explained as Sim1 may be related to large feather formation (probably muscle formation). While Sim 1 may be important, its role may be a consequence, or supportive, not casual for flight trait evolution. They should characterize those Sim1 positive cells more, and what they become.

5. The targeted mutation or deletion of the reporter sequence would be needed to validate the expression experiment.

6. It would be helpful if authors could show the ChIP-seq data to demonstrate the differential regulation of Sim1 between forelimb and hindlimb. But this point is optional.

7. Authors can do more on Inadl, Boc, Pax9. May be they also provide some clues to the big question.

Reviewer #3 (Remarks to the Author)

Summary: in this manuscript the authors investigate the genetic mechanisms underlying the macroevolutionary transition from non-avian dinosaurs to birds. By analyzing 48 avian genomes and comparing with 9 non-avian vertebrate genomes, the authors identify millions of avian specific highly conserved elements (ASHCEs) and find that these are predominantly located in non-coding regions. Analysis of polymorphism rate within chicken populations suggests that ASHCEs are also at recent under strong selective constraints. The authors also analyze the chromatin-state landscape of ASHCEs in several stages of chick embryo development and also in limb buds and show that over 25% of ASHCEs are within histone peaks. Through different analyses, the data also show that ASHCE-associated genes are specifically active in late avian embryonic stages supporting their involvement in developing avian specific features. The authors concentrate in the limb and perform large-scale in situ comparative embryonic expression analysis for genes with most highly conserved ASHCEs. Finally the authors concentrate in the transcription factor Sim1 for which they determine the expression pattern in chick, gecko and mouse and the enhancer activity of its associated ASHCE in chick and mouse.

Critique: The manuscript is well written, and in particular, the introduction does a good job of setting the stage for the reader. The data is interesting and thorough and the amount of work performed by the authors absolutely astonishing. I find a little unclear which data has been generated by the authors and which used from data repositories. For example, did they sequence the genome of a population of chicken for the polymorphism analysis or for the generation of the gene expression-profiling map? I also think that the rationale behind some of the analysis could be better explained. Why did the authors select the turtle to compare the expression levels of chicken and other non-avian outgroups? If only 13 genes are differentially expressed at later stages between chick and turtle, why didn't the authors comment on these 13 genes?

Overall, this manuscript addresses a relevant issue and integrates analytical genomics, developmental biology and paleontology analyses. I think it provides compelling evidence of class-specific regulatory elements highly conserved in avian genomes and their preferential location in non-coding sequences,

Minor point-It seems to be some disconnection between the text and the figures. For example, in Fig1 the ASHCEs are classified in type I and II but this terminology is not mentioned in the text. Also, in Fig2e the region upstream DLG1 is used as example of the temporal changes in histone marks but this is not mentioned in the text.

One by one responses to the Reviewers' comments:

Reviewer #1 (Remarks to the Author):

A. Summary of the key results

- Seki et al. identify avian-specific highly conserved elements (ASHCEs) and ask if the genes they are associated with show evidence of being involved in specifying avian-specific morphological features.
- They first identify ASHCEs using a comparative genomic approach and show that they are depleted in coding regions compared to HCEs
- They use chromatin immunoprecipitation experiments, publically available whole embryo chick RNA-seq time-course data, and molecular evolutionary analyses, to guide a large scale comparative in situ hybridization strategy that seeks to discover genes whose expression has been rewired during avian evolution. Their study focused on limb-related phenotypes.
- Results from in situ hybridization analyses highlight several genes that appear to have a distinct expression in chick compared to mouse and gecko.
- The authors focus on one transcription factor Sim1 and show that its associated ASHCE can drive a unique gene expression pattern in both chick and mouse embryos.

Response: Thank you for your time spent on reviewing our manuscript.

B. Originality and interest: if not novel, please give references

- This is an original and interesting study. Cotney et al. Cell 2013 (PMID:

23827682) have previously used comparative epigenetic analyses of H3K27ac to discover lineage specific limb enhancers in primates. Infante et al. Dev Cell 2015 (PMID: 26439399) have recently looked for limb-genital enhancer usage in snakes, again relying on H3K27ac to identify such enhancers. However this study appears to be the first genome-scale effort to explore this important question in birds.

Response: We sincerely appreciate your high evaluation of our findings. We have further added new findings in this revision, and we hope these would fully answer to your comments.

C. Data & methodology: validity of approach, quality of data, quality of presentation

- Although they gave a link to their data, the authors did not provide a password for the reviewer to access their data. Given the detailed QC and informative summary given by the authors I presumed the data was indeed submitted to GEO. I recommend that the authors consider providing the many processed data files (e.g. entire list of ASHCEs, genome browser tracks and differential enrichment analyses of ChIP-seq data, list of top500 genes), to increase the impact and value of their study to the research community.

Response: We apologize that we omitted the link to the data. The raw reads, predicted peaks and normalized signal files of ChIP-seq data in BedGraph format (can be easily loaded into genome browsers) were uploaded to GEO (accession GSE75480, see the link below for reviewing). Other processed files are provided in Supplementary Tables (see

Supplementary tables 37-49) in the revised manuscript.

Link for reviewing the GEO data:

<http://www.ncbi.nlm.nih.gov/geo/query/acc.cgi?token=edkjkqalbexhyl&acc=GSE75480>

- Figure 1 has some minor grammatical issues in figure and legend.

Response: We have checked and corrected grammatical issue.

- I find that "47% of ASHCEs contained at least one transcription factor binding site (TFBS)." is not really that compelling. I presume almost any sequence would have at least one predicted TFBS. It may be leveraging the epigenetic datasets collected in this study prior to doing motif enrichments (i.e. do limb enriched H3K27ac regions give any specific motif enrichments?).

Response: Following the reviewer's suggestion, we did motif discovery based on the data of histone ChIP-seq peaks to identify additional putative TFBSs, and re-did the enrichment analysis with ASHCEs data set (Supplementary Table 9). Plus the previously identified TFBSs, we found that the 58.33% of ASHCE have at least one TFBS.

We identified 107 over-represented TFBSs in the limb-specific differential sites overlapping ASHCEs, and summarized this data in Supplementary Table 23.

- On page 8 the identity of the lincRNAs is not given. Only reference I could find appear to be analysis-specific annotations in the supplemental file (e.g. "CUFF.3981"). I recommend the authors provide a list of the differentially expressed genes used in their analyses.

Response: We provided the GFF annotation file of lincRNAs and the list of differentially expressed lincRNAs as Supplementary Tables (see Supplementary Tables 43-44) in the revised manuscript.

- On line 169 authors mention they created 'a chromatin state map'. No chromatin state map is mentioned in the methods nor is one shown in Figure 2b. To me chromatin state maps are used to integrate different histone modifications (i.e. PMID: 23221638). Such an analysis of the whole embryo and limb ChIP-seq data could be useful to compare to ASHCEs.

Response: We apologize for the confusing description. The original meaning of "chromatin state map" in the manuscript was referring to the genome-wide histone ChIP-seq profiles generated in this project, rather than the genome segmentation maps predicted by methods like ChromHMM/Segway. Based on the reviewer's suggestion, we have generated the 4-state chromatin maps predicted by ChromHMM and added them to our revised manuscript (see description at section on 'Chromatin-state landscape of ASHCEs' in main text and Supplementary Figure 2 & Tables 18-20). According to the combinatorial patterns of the three marks (Supplementary Figure 2) and observations of previous

literature, we roughly defined the four states as “strong enhancer”, “weak enhancer”, “poised enhancer” and “low signal”. However, because we only have 3 types of histone marks and the combinatory code of histone modifications in birds could be different from that in other previously investigated species (e.g., mammals), the conclusion from predicted state maps analyses might not be that strong. Also, due to the limited space, we organized this result in supplementary information. For example, the *DLG1* promoter region (see the figure below) has strong signal of H3K4me1 and H3K27ac and weak signal of H3K27me3, but ChromHMM predicted this region to be a strong enhancer. Because H3K4me1/H3K27ac/H3K27me3 can also be found in regions other than enhancers (e.g. promoters), the predicted chromatin state maps based solely on the three marks have relatively low accuracy and resolution and therefore we are cautious in inferring too much from these maps.

- Summary Lines 173-176 - the results reported for the ChIP-seq experiments appears to be more of a QC analysis. Could the authors classify all their ASHCEs based on some epigenetic criteria (e.g high H3K27ac, low H3K4me1 = putative promoter; high H3K27ac, low H3K27me3 active enhancer; high H3K27me3, high H3K4me1 latent/poised enhancer)? For example there is no mention of which ASHCEs show evidence of H3K27ac enrichment (in whole embryo or limb). Also the biology of H3K27me3 is not brought into the study. Do any ASHCEs show evidence being regulated by polycomb complex? Were any H3K4me1+H3K27me3 high and H3K27ac low regions found amongst the ASHCEs? Are there any ASHCEs that are also H3K27ac enriched in limb? If so I would recommend including/discussing these regions in the later in situ experiments. I would argue that an epigenetically active ASHCE in developing chick limb is more biologically relevant to limb development than an ASHCE ranked by the DNA constraint of its closest gene. The H3K27me3 data generated in this study could result in unique discoveries, however it seems to me this data was not effectively used to explore the biology of ASHCEs.

Response: We agree that many other interesting points can be explored with the ChIP-seq data. However, the focus of this study is ASHCEs and their potential functional roles, and we regret that other things are beyond our capacity (especially within a limited time for the revision). The main purpose of ChIP-seq data generated in the study was to see whether ASHCEs are enriched with enhancers-associated histone marks. After generating the 4-state chromatin maps, we classified the ASHCEs

into different states in limb/whole embryo samples of each surveyed developmental stage (Supplementary Table 18), and found that two states, "strong enhancer" and "weak enhancer", are over-represented at all the surveyed stages (Supplementary Table 19). We appreciate the interest and power to identify candidate functional regions with histone modification. However, our main focus in this manuscript is to explore potential functions of ASHCEs. Therefore, we have mainly used conservation score to prioritize the candidate genes. We also agree with the reviewer that combination of ASHCEs and the ChIP-seq data could identify more genes that are important for development. We followed this suggestion and identify a list of genes/regions with epigenetically active ASHCEs (see Supplementary Figure 3 and Supplementary Tables 22,48-49). These genes could be useful for the further experiment.

- Figure 3A is cluttered and how the annotations are placed on the figure is not clear. I am not sure this is a helpful addition to the main text of this manuscript.

Response: The purpose of Fig. 3A is to highlight the enrichment of development-related GO terms in the top 500 ASHCE-associated genes. Since the list of enriched GOs was long, the visualization tool REVIGO (<http://revigo.irb.hr/>) was used to cluster them based on semantic similarity and highlight the development-related GOs. In the revised manuscript, we modified the legend as follows:

"a. Enriched GO terms in the top 500 ASHCE-associated genes. The p-values of enrichment were calculated using a chi-squared test, and FDRs were computed to adjust for multiple testing. Since the list of enriched GOs was long, the figure was generated by the visualization tool

REVIGO which clustered the GOs based on semantic similarity. The development-related GOs are highlighted with bold fonts”.

- On line 187 the authors report: "We also found that some ASHCEs harbored with sites showing significantly up-regulated histone modification in limb samples compared to whole embryo samples. These ASHCEs might be associated with the limb-specific enhancer functions (Supplementary Table 19)." The main text and Table 19 legend seem to imply that there are 564,596 limb-specific enhancers in ASHCEs. However there are only 34,000 limb enhancers in total reported for the HH32 limb ChIP-seq data. This needs to be clarified. It may be the authors are referring to the number of base pairs of ASHCEs that are overlapping H3K27ac enriched peaks. However the text and supplemental figure legend seem to refer to ASHCEs themselves, not individual base pair coverage of ASHCEs. Without access to the processed data I am not able to look further into this. This is crucial as it is essentially the only analysis that the authors perform where individual ASHCEs are compared to the epigenetic data.

Response: Thank you for spotting this. Indeed, the numbers in Supplementary Table 19 represent the base pairs of ASHCEs that are overlapping ChIP-seq peaks. We added the information of units (bp) to the table in the revised manuscript.

The previous limb enhancer evolution studies mentioned earlier, underscore the rationale for leveraging epigenetic data in discovering

limb enhancers. More effort to integrate/compare the epigenetic results to the DNA constraint-based analyses would likely provide important biological insights.

Response: We did additional analyses on the ChIP-seq data and the results have been added to the revised manuscript. As stated above, we generated the chromatin state maps using chromHMM and connected the new data to ASHCEs (see Supplementary Figure 2 & Tables 18-20). Regarding limb related genes, we analyzed the 16 limb related genes (based on GO annotation) in the top 500 list and plotted the ChIP-seq tracks of five genes (*PRRX1*, *LEF1*, *MEOX2*, *GLI3*, *FMN1*) which show differential histone modification between limb and whole embryo samples in the ASHCE regions (Supplementary Figure 3).

- The explanation for the 'differential histone modification' in Figure 2e is incomplete. For example which of the 3 histones are differentially enriched (and at what cutoff). It appears there may be a decrease in H3K27ac, H3Kme1 at HH32 and an increase in H3K27me3 relative to HH21.

Response: We added more explanation to the legend of figure 2e. We also indicated the differential regions between HH21 and HH32 predicted by diffReps in the figure. The legend is modified as below:

“(e). A case of differential histone modification marks in the upstream of *DLG1*. Normalized fold enrichment signals (normalized with input samples) of three histone marks at HH21 and HH32 stages are shown. The differential regions between HH21 and HH32 predicted by diffReps

are also shown (bars under HH32 tracks)".

The fact that H3K27me3 is a repressive histone modification is not mentioned in the manuscript. A more thorough analysis of the histone modification data and a clear linking of the results to the existing gene expression data and novel in situ data would be helpful. For example the only example given was at DLG1, however I did not see any discussion of this gene later in the manuscript. I was wondering why it was chosen as opposed to the ASHCEs explored later in the paper? In principle a detailed analysis of the epigenetic data will provided classes of ASHCEs and the authors could provide examples of representative ones. For example what is the epigenetic status of Sim1 in limb (the epigenetic data would fit well as part of Fig 5a for example)? It is of course possible that the avian specific innovations occur in a small number of cells and that bulk ChIP-seq or RNA-seq will not have the power to detect them. If nothing was learned from the epigenetic status of ASHCEs in the developing limb then this needs to be made clear as it would contradict results from previous studies. Such ideas/caveats should be explored in the paper.

Response: DLG1 was a development-related gene selected to show the dynamic changes of histone modification in ASHCEs during chicken development. We added a sentence describing DLG1 in the revised text: "For example, an upstream ASHCE of the gene DLG1, which is found involved in embryo development in mouse, exhibits down-regulated H3K4me1/H3K27ac and up-regulated H3K27me3 at the HH32 stage compared to HH21 (Fig. 2e), suggesting a transition of the underlying

regulatory function”.

Regarding the epigenetic status of Sim1 in limbs, we did check the ChIP-seq data around the Sim1 gene region (see the figure below), but did not find significant difference between stages for the ASHCE (the ASHCE in the dash-line box of the figure), probably due to only a small number of cells expressing Sim1 in the flight feather region. Although the epigenetic status of this gene does not change significantly between samples, we did observe a unique expression pattern of this gene.

- Line 217: clarify which prediction is being referred to.

Response: Thank you for the correction. We re-wrote this sentence.

- It is not clear how the top100/500 list used for the comparative gene expression in situ analyses (line 235) connects to the ChIP-seq experiments. This should be made clear. Based on the previous limb enhancer literature I would be particularly interested in the function of ASHCEs near known limb/genital enhancers. However the sense I get from this paper is that DNA sequence constraint was the primary means of prioritizing limb enhancer innovations.

Response: Since the main purpose of this study is to explore the potential function of ASHCEs, we mainly used conservation score to guide the candidate gene selection. But we did not exclude that the change of epigenetic regulation could also be another mechanism important to the limb development. Following this suggestion, we analyzed the 16 limb related genes (based on GO annotation) in the top 500 list and provided the ChIP-seq tracks of five genes (PRRX1, LEF1, MEOX2, GLI3, FMN1) which show differential histone modification between limb and whole embryo samples in ASHCE regions (see Supplementary Figure 3).

- Line 255 uses Indl abbreviation and rest of manuscript uses Inadl.

Response: Thank you for the correction. We have corrected it.

- Although the QC provided by the authors is well described, it is not clear how robust the ChIP-seq experiments are. Is the y axis in Figure 2E some sort of normalized value or does it represent read number. It would be helpful to show input on the same scale. More examples of genome browser tracks for the most important, and representative ASHCEs would be valuable.

Response: The y axis in Fig. 2e represents the normalized fold enrichment signals by comparing ChIP-seq data against input samples (we clarified this in the revised legend). In the original manuscript the assessments of robustness were provided in the methods and Supplementary Tables 12-14 (eg. measures such as NSC, RSC and Pearson correlation), and the same tables can be found in the revised manuscript. We also added more examples of genes of interest with histone ChIP-seq tracks in the revised manuscript (see Supplementary Figure 3).

- Figures 5B, C and D are somewhat difficult to interpret as the results presented here do not seem to show consistent EGFP expression within the Sim1 endogenous expression domain in the chick embryo. In contrast the mouse reporter data looks more convincing. It seems more work is needed to justify the author's conclusions in the chick system. For example, a previous study that identified Sim1 as a chick and mouse limb enhancer (PMID: 19123137; this paper was not cited in the current study) mention that they had to resort to doing in situ work sections.

Response: Fig. 5b and c are based on electroporation. This method, which is one of the best technologies for reporter assay at late stages of organ development in avian embryos, always results in mosaic transfection. Like other studies, we cannot avoid this situation whenever using this method. In addition, in order to stably express reporter GFP in the later stages, not only a GFP plasmid but also a transposase one is required (this mediates genomic integration of GFP) to be introduced together into the same cells (see Sato et al. (Dev. Biol., 2007, 305, 616-624) for detail). This is probably why the small number of cells are GFP-positive in our experiments. We performed this experiment repeatedly and robustly confirm the expression pattern of Sim1. All of the data are summarized in Supplementary Figure 10.

In Fig. 5d, positive signals can be seen in the endogenous Sim1-expressing region, although the signal intensity is a bit weak. The relatively-low signal intensity could be due to TK minimal promoter used here.

We answer the question about expression pattern of Sim1 gene below in more detail.

- Supplemental Figure 2 seems to be blurry compared to supplemental Figure 3.

Response: We provided a higher-resolution version for Supplemental Figure 2 in the revised manuscript.

D Appropriate use of statistics and treatment of uncertainties

- No caveats are given regarding the limitations of using of whole embryo RNA-seq data to identify ASHCE-genes in development (line 207).

Response: We apologize that we might have misled the reviewer. We did not identify any candidate gene based on RNA-seq data, but we simply investigated when these ASHCE-associated genes get expressed during embryogenesis. Having said so, it is very much to the point that it is better to clearly mention the limitations of the analysis.

We have modified some of our sentence, including the following, in the result section (see 1st paragraph of page 12, lines 256-259) and added the limitation of this method:

“,,, However, whole embryonic RNA-seq dataset we used in these analyses missing the anatomical information (e.g. the regions where genes are expressed), and we decided to further investigate the role of ASHCE-associated genes at tissue level.”

- How variability in the in situ work was treated was not clear. How many embryos were analysed for each in situ and what criteria was used to select a representative pattern could be made more clear.

Response: According to this comment, we added some description in “Methods” section as follows:

In expression screenings in chicken and mouse embryos (Supplementary Figs. 4, 5), we used around three (at least two) individuals for each gene

at each developmental stage. Then, we confirmed that the candidate genes showed almost consistent expression pattern in all specimens we analyzed. Regarding the expression analysis in gecko (Supplementary Fig. 6), we used one embryo for each gene and developmental stage because of a limited number of the fertilized eggs for this animal

E Conclusions: robustness, validity, reliability

- The identification of the ASHCEs appears to be robust.

F Suggested improvements: experiments, data for possible revision

- Gene expression data (qPCR or RNA-seq) that matches the limb development ChIP-seq data would be ideal and could allow authors to better identify genes which are regulated during limb development.

Response: We agree with the reviewer that it is interesting to explore ChIP-seq data together with RNA-seq data in order to understand the limb development. As replied above, we have provided some preliminary analyses on ChIP-seq data and a list of candidate genes associated with ChIP-seq status change. However, since our main research focus here is on the potential function of ASHCEs, the detail exploration on ChIP-seq data is out of the research scope in this project. We agree that the data and the candidate genes would be useful for the further study as indicated by the reviewer.

- In their last paragraph of the section "Functional roles of ASHCE-genes in development", the authors identified "13 ASHCE-associated genes" that "might be the candidates with specific function in chicken"

from comparison of chicken and turtle. I am curious to know what the expression patterns of these 13 genes in their in-situ analysis. Do they show unique expression pattern in chicken? Are they among the top 100 ASHCE-associated genes?

Response: While we agree with the reviewer that it is possible that these 13 gene candidates may have unique and specific tissue expression pattern in chicken, the RNA-seq data for the chicken and turtle comparison was from whole embryo samples and could not tell the tissue expression pattern. Our current study only focused on the limb development, thus only did the in-situ hybridization in limb. But the detail expression patterns of these 13 genes could be useful for further experiment studies.

- (line 205) The most conserved genes (top 500) nearby an ASHCE's were said to be enriched for "limb development". However the genes that gave rise to this enrichment were not mentioned (Table 21 mentions there are 16 of them. Note Table 23 was referenced in place of Table 21 where the enrichment was given). It is unclear if any of the known limb genes that had a nearby ASHCE were explored or highlighted in the ChIP-seq, RNA-seq, or in situ experiments. Is Sim1 one of these known genes? For these known limb genes it would be justified to identify all nearby ASHCEs and not be restricted to only 10kb upstream and down stream. Epigenetic activation in the limb could also be used to prioritize the functional study of these limb related ASHCEs. Given the tremendous effort already put into in situ hybridization it would be worth exploring this angle and perhaps even profiling a few of these if they haven't already been done.

Response: Sim1 was not annotated with GO term under 'limb development'. The genes enriched for 'limb development' were listed at Supplementary Table 25. We investigated the ChIP-seq data of the 16 limb-related genes and found five of these genes whose associated ASHCEs showed differential histone marks. We displayed the ChIP-seq tracks for these five genes in the revised manuscript (see Supplementary Figure 3). Our study only selected the 100 genes from top 500 gene list for in-situ hybridization experiment. Note that genes in the top 500 list with poor annotation were not considered (This has been clarified in the methods section) and we didn't restrict the selected genes to those with known developmental functions. Two (PRRX1 and LEF1) of the 16 'limb development' genes were included in the 100 genes selected for in situ experiments, but others were not.

- It is understandable why the authors focused on in situ patterns that were different in chick and mouse. However the recent literature (e.g. Cotney et al. and Infante et al.) seem to suggest to rewiring of existing limb genes has lead to lineage-specific morphological innovations. If it is difficult to assess the quantitative differences of limb gene expression via in situ hybridizations between species, would it not be reasonable to prioritize all limb related genes with a nearby ASHCE rather than using DNA constraint scores for prioritization? For example a previous study that showed Sim1 was a chick limb enhancer concluded the expression patterns were similar between chick and mouse (PMID: 19123137). To me the expression patterns driven by the enhancer associated with TFAP2C seems to be another excellent candidate. The authors have done a huge amount of in situ validations however more discussion of

the current literature and the biology behind alternative prioritization schemes (e.g. known limb genes, epigenetic activity as opposed to measures of purifying selection may have been a more appropriate way to identify limb related enhancers.)

Response: We fully agree that using other strategies, like ChIP-seq data and the annotated limb-related genes, could also provide powerful ways to identify important genes function in limb development. However, the main purpose of this study is to investigate the overall functions of ASHCEs. Our experiments did show that the list of ASHCEs-associated genes that we identified was an important dataset for functional study. Some of these genes are overlapping with the candidate genes that were identified from other strategies. Therefore, a combination of all strategies would be very useful to target on the limb development genes. We agree that TFAP2C could be another nice candidate gene for functional examination. It would be worth to explore its function in further study.

Sim1 expression can be seen in the relatively early stages of limb development both in chick and mouse. This is because Sim1 is expressed in the developing and migrating muscle precursor cells in the limb bud as the paper (PMID: 19123137) reported. Our experiment also confirmed this (Supplementary Fig. 6e). However, our experiment also detected the specific expression of Sim1 in the feather bud-forming region of chicken embryo at a later stage. This is the unique pattern in chicken and not shows in mouse and gecko.

We revised the part of the main text and cited this reference as follows:

“,,, whereas similar expression patterns in all three species could be seen around the basal region of the hindlimb (Fig. 4a) and other areas such as the somite and migrating muscle precursors migrating into the limb bud 34 (Supplementary Fig. 6e)

G References: appropriate credit to previous work?

- The authors have not acknowledged or incorporated ideas from papers directly related to their work. For example:

1) Limb enhancer study of Cotney et al. How does this work relate to the current study?

Response: Thank you for this comment. The study by Cotney et al. provided an excellent dataset of potential enhancers and promoters in limb development. It also showed a wonderful example of using histone modification profiling to identify functional regions. We have cited this paper in our revision (see ref. 21).

2) Infante et al. Is there evidence of genital expression patterns changing along with limb expression patterns during avian evolution?

Response: As some reports suggested similarity of developmental mechanism between limb and genitalia, it is possible that the avian genital organ has specific expression pattern similar to the limb bud.

We, however, do not find any evidence in this study to show this. It could be interesting to explore in a future study.

3) The association with Sim1 to limb development in chick and mouse seems to have already been published in 2009 (PMID: 19123137). The early work claimed that Sim1 had a similar expression in chick and mouse which is different than the conclusions of this study. A comparison of this previous work to the current work is warranted.

Response: Please excuse our poor description on this point. Yes, Sim1 expression can be seen in the relatively early stages of limb development both in chick and mouse. This is because Sim1 is expressed in the developing and migrating muscle precursor cells in the limb bud as the paper (PMID: 19123137) reported. Our experiment also confirmed this (Supplementary Fig. 6e). However, our experiment also detected the specific expression of Sim1 in the feather bud-forming region of chicken embryo. This is the unique pattern in chicken and not shows in mouse and gecko.

We revised the part of the main text and cited this reference as follows: ",,, whereas similar expression patterns in all three species could be seen around the basal region of the hindlimb (Fig. 4a) and other areas such as migrating muscle precursors migrating into the limb bud 34 (Supplementary Fig. 6e)

H Clarity and context: lucidity of abstract/summary, appropriateness of

abstract, introduction and conclusion

- The context and rationale are compelling and clear. The relationship between different experiments/analyses in the paper could be strengthened as mentioned above.

Response: We highly appreciate this reviewer's great comments, which have improved our manuscript nicely.

Reviewer #2 (Remarks to the Author):

Authors ask a great question on the macro-evolutionary process from reptiles to the birds. In an attempt to identify genes responsible for avian-specific features, Seki et al performed a comparison of the 48 avian genomes with 9 non-avian species looking for avian specific highly conserved elements (ASHCEs). They found that the vast majority of ASHCEs were located in cis regulatory non-coding regions rather than producing novel coding genes. Using an extensive in situ hybridization screen covering different developmental stages to identify expression patterns, they identified several genes that correlated with bird-specific limb development, including bone and feather formation. Four of these genes (*Inadl*, *Boc*, *Pax9* and *Sim1*) were not present in mouse or gecko limbs and were further characterized. The expression of *Sim1* in the chicken limb suggests it may be involved in the acquisition of flight feathers. This result was tested using electroporation of a reporter construct, as well as transduction with a retroviral construct and the use of transgenic mice. *Sim1* ASHCE directed reporter expression in the location similar to the endogenous gene. While they

have done a body of works that identify many cis-regulatory elements, and increase our knowledge toward the wonderful big question asked, it is not convinced that they have found genes that define "bird specific trait" or define the avian class.

1. Coumailleau and Duprez, *Int. J. Dev. Biol.* 53:149-157 (2009) using section in situ hybridization reports that Sim1 shows similar expression patterns in both mice and chickens at similar stages as described here. They demonstrate that it is involved in early forelimb muscle formation. This paper was not mentioned in the current manuscript. How do the authors reconcile this paper with their own findings?

Response: Please excuse our poor description on this point. Yes, Sim1 expression can be seen in the relatively early stages of limb development both in chick and mouse. This is because Sim1 is expressed in the developing and migrating muscle precursor cells in the limb bud as the paper (PMID: 19123137) reported. Our experiment also confirmed this (Supplementary Fig. 6e). However, our experiment also detected the specific expression of Sim1 in the feather bud-forming region of chicken embryo at a later stage. This is the unique pattern in chicken and not shows in mouse and gecko.

We revised the part of the main text and cited this reference as follows:
",,, whereas similar expression patterns in all three species could be seen around the basal region of the hindlimb (Fig. 4a) and other areas such as the somite and migrating muscle precursors migrating into the limb bud 34 (Supplementary Fig. 6e)

2. Although they show ASHCE found in Sim1 can direct expression to the specific region of the limb adjacent to where flight feathers form, they don't clearly show that Sim1 is involved in limb or feather development. Functional experiment is essential to establish the causal relationship. If Sim1 is a real flight feather determination gene, overexpression of Sim1 ASHCE or Sim1 gene body in the specific region in hind limb should induce the flight feather formation in the leg. The authors should do this experiment.

Response: We did not intend to claim that the Sim1 gene was a flight feather determination gene. Our reporter assay experiment, gene expression pattern of Sim1 in chicken limb and tail, and the evolutionary analysis all indicate its association with flight feather evolution.

We followed the next comment on the flight feather in legs, and succeeded to obtain some embryos of two chicken strains with flight feathers in the leg. As shown in new Figure 4k&l, and Supplementary Figure 8, we also detected a clear expression of Sim1 in the posterior margin of the leg autopod, where the ectopic flight feathers develop in these two chicken strains. This expression pattern was never seen in the wild type chicken. This new result provides another strong evidence that Sim1 is involved in flight feather formation rather than it is randomly expressed in the posterior margin of the wing bud.

3. They should also evaluate the expression of Sim 1 in the hind limbs of birds with feathers on their feet. Shapiro claims that these are flight feathers and authors might expect to see Sim1 expressed with a similar

pattern as that seen in forelimbs in these birds as opposed to birds with scales on their feet.

Response: Thank you for this great comment. And please see our above response. We performed this experiment and confirmed the expression of Sim1 in the flight feather leg.

4. Sim1 expressed in both wing feather and tail feather. The expression of Sim1 in these feathers could be explained as Sim1 may be related to large feather formation (probably muscle formation). While Sim 1 may be important, its role may be a consequence, or supportive, not casual for flight trait evolution. They should characterize those Sim1 positive cells more, and what they become.

Response: Thank you for this critical comment. As mentioned in the original and revised manuscript, Sim1-positive cells are neither myogenic, epidermal nor nervous. We are also curious if Sim1 was a cause or consequence of flight trait evolution. However, it is far beyond our scope to definitively clarify the causality of Sim1 in the evolution of avian flight traits (The main focus of our paper is potential function of ASHCEs). And not to mention the imperfection of the concept 'causality' (as discussed by David Hume), we do not think clarifying the characteristics of Sim1 positive cells is sufficient for answering the above question (though would be suggestive for flight feather 'development'). Having said so, we should avoid potential over-statements that make readers to misunderstand, and changed our abstract as follows.

"In particular, we pinpointed the avian-specific expression of gene Sim1

driven by ASHCE may be associated with evolution and development of flight feather.”

5. The targeted mutation or deletion of the reporter sequence would be needed to validate the expression experiment.

Response: Although the targeted mutation or deletion could add another evidence to support the function of reporter sequence, it will not add any other new insight into our conclusion that this ASHCE has regulatory function to Sim1 gene. We believe that the reporter assay experiments in both chicken and mouse embryos provide sufficient evidence to support this.

6. It would be helpful if authors could show the ChIP-seq data to demonstrate the differential regulation of Sim1 between forelimb and hindlimb. But this point is optional.

Response: This is an interesting point. But since this experiment is not related to our main topic on the function of ASHCEs and the limb samples used in our ChIP-seq analysis were mixtures of fore- and hindlimbs, we prefer to consider it in a future study.

7. Authors can do more on Inadl, Boc, Pax9. May be they also provide some clues to the big question.

Response: Thanks for the reviewer’s appreciation on the candidate

genes we found. Our main focus in this manuscript is to explore the potential function of ASHCEs. But we agree that it could be interesting for further study to test the functions of other genes in our list.

To tone down this point and focus Sim1 as an example, we moved the pictures of Inadl, Boc and Pax9 from the main Figure 4 to Supplementary Figure 7.

Reviewer #3 (Remarks to the Author):

Summary: in this manuscript the authors investigate the genetic mechanisms underlying the macroevolutionary transition from non-avian dinosaurs to birds. By analyzing 48 avian genomes and comparing with 9 non-avian vertebrate genomes, the authors identify millions of avian specific highly conserved elements (ASHCEs) and find that these are predominantly located in non-coding regions. Analysis of polymorphism rate within chicken populations suggests that ASHCEs are also at recent under strong selective constraints. The authors also analyze the chromatin-state landscape of ASHCEs in several stages of chick embryo development and also in limb buds and show that over 25% of ASHCEs are within histone peaks. Through different analyses, the data also show that ASHCE-associated genes are specifically active in late avian embryonic stages supporting their involvement in developing avian specific features. The authors concentrate in the limb and perform large-scale in situ comparative embryonic expression analysis for genes with most highly conserved ASHCEs. Finally the authors concentrate in the transcription factor Sim1 for which they

determine the expression pattern in chick, gecko and mouse and the enhancer activity of its associated ASHCE in chick and mouse.

Critique: The manuscript is well written, and in particular, the introduction does a good job of setting the stage for the reader. The data is interesting and thorough and the amount of work performed by the authors absolutely astonishing. I find a little unclear which data has been generated by the authors and which used from data repositories. For example, did they sequence the genome of a population of chicken for the polymorphism analysis or for the generation of the gene expression-profiling map? I also think that the rationale behind some of the analysis could be better explained. Why did the authors select the turtle to compare the expression levels of chicken and other non-avian outgroups? If only 13 genes are differentially expressed at later stages between chick and turtle, why didn't the authors comment on these 13 genes?

Response: We mentioned in the main text that the population data was from previously study (Wong et al 2004) and cited the ref. We selected turtle as an out-group because turtle (*P. sinensis*) is the closest non-avian species that both has had its genome sequenced (Wang et al 2014) and has developmental transcriptome data available for genome-wide comparison. We have added the list of these 13 genes in Supplementary Table (see Supplementary Table 34). However, due to the limited space, we did not discuss these genes in more detail in the main text.

Overall, this manuscript addresses a relevant issue and integrates

analytical genomics, developmental biology and paleontology analyses. I think it provides compelling evidence of class-specific regulatory elements highly conserved in avian genomes and their preferential location in non-coding sequences,

Minor point-It seems to be some disconnection between the text and the figures. For example, in Fig1 the ASHCEs are classified in type I and II but this terminology is not mentioned in the text. Also, in Fig2e the region upstream DLG1 is used as example of the temporal changes in histone marks but this is not mentioned in the text.

Response: We have added more detailed explanation for both Figs 1 and 2 in main text.

Reviewers' Comments:

Reviewer #1 (Remarks to the Author)

The authors took advantage of the reviewers comments and the revisions and new data have improved this already interesting work.

Reviewer #2 (Remarks to the Author)

In the abstract, authors state many ASHCEs show differential histone modifications. However, all of Sim1 associated ASHCEs do not show differential histone modifications.

Authors explain that the differential histone modification signals of Sim1 could be averaged out in their mixed tissues. Another possible explanation is that Sim1 enhancers could be far away from the region they explored (only 10 kb upstream and downstream of Sim1). This should be stated in the discussion.

REVIEWERS' COMMENTS:

Reviewer #1 (Remarks to the Author):

The authors took advantage of the reviewers comments and the revisions and new data have improved this already interesting work.

Response: Thanks to the reviewers for all the insightful comments, which helped improve the manuscript.

Reviewer #2 (Remarks to the Author):

In the abstract, authors state many ASHCEs show differential histone modifications. However, all of Sim1 associated ASHCEs do not show differential histone modifications.

Authors explain that the differential histone modification signals of Sim1 could be averaged out in their mixed tissues. Another possible explanation is that Sim1 enhancers could be far away from the region they explored (only 10 kb upstream and downstream of Sim1). This should be stated in the discussion.

Response: In our manuscript we have provided strong evidence supporting that the ASHCE in a Sim1 intron exhibits enhancer activities in chicken. Although there could be other Sim1 enhancers that could be far away from the region we explored, we don't think it is necessary to add this to the discussion because it is not very relevant.